# Supporting Electrification Policy in Fragile States: A Conflict-Adjusted Geospatial Least Cost Approach for Afghanistan

**Alexandros Korkovelos [1,\*], Dimitrios Mentis [1,2], Morgan Bazilian [1,3], Mark Howells [4], Anwar Saraj [5], Sulaiman Fayez Hotaki [5] and Fanny Missfeldt-Ringius [6]**

[1] Division of Energy Systems, KTH Royal Institute of Technology, Brinellvägen 68, 100 44 Stockholm, Sweden; Dimitrios.Mentis@wri.org (D.M.); mbazilian@mines.edu (M.B.)

[2] World Resources Institute, Washington, DC 20002, USA

[3] Colorado School of Mines, 1500 Illinois St., Golden, CO 80 401, USA

[4] Department of Geography, Climate Compatible Growth (CCG) Program, Loughborough University, Epinal Way, Loughborough LE11 3TU, Leicestershire, UK; M.I.Howells@lboro.ac.uk

[5] Geomatics and Engineering Department, Kabul Polytechnic University, Bagh-e-Bala Road, 5th District, Kabul 1010, Afghanistan; anwarsaraj@gmail.com (A.S.); sf.hotaki@kpu.edu.af (S.F.H.)

[6] The World Bank Group, Washington, DC 40433, USA; fmissfeldt@worldbank.org

\* Correspondence: alekor@kth.se; Tel.: +46-(0)735-84-36-13

**Abstract:** Roughly two billion people live in areas that regularly suffer from conflict, violence, and instability. Infrastructure development in those areas is very difficult to implement and fund. As an example, electrification systems face major challenges such as ensuring the security of the workforce or reliability of power supply. This paper presents electrification results from an explorative methodology, where the costs and risks of conflict are explicitly considered in a geo-spatial, least cost electrification model. Discount factor and risk premium adjustments are introduced per technology and location in order to examine changes in electrification outlooks in Afghanistan. Findings indicate that the cost optimal electrification mix is very sensitive to the local context; yet, certain patterns emerge. Urban populations create a strong consumer base for grid electricity, in some cases even under higher risk. For peri-urban and rural areas, electrification options are more sensitive to conflict-induced risk variation. In this paper, we identify these inflection points, quantify key decision parameters, and present policy recommendations for universal electrification of Afghanistan by 2030.

**Keywords:** Afghanistan; conflict; geospatial electrification; OnSSET; Geographic Information Systems (GIS)

## 1. Introduction

Access to affordable, reliable, and modern energy services has been recognized as a key enabler of United Nations' sustainable development goals (SDGs) [1]. Electrification, in particular, has long being emphasized to "power" opportunities for socioeconomic uplift, growth, and well-being in least developed areas [2–4]. Despite progress, nationwide electricity access by 2030 is still an ambitious goal for many countries [5]. It requires the motivation of significant investment in usually highly uncertain environments, e.g., poor, rural settings in least developed or fragile areas. Consequently, these areas usually rank quite low in terms of modern service delivery. Yet, it is where people are most vulnerable and in high need of that service.

The World Bank estimates that two billion people live in areas where the state of safety and security is fragile [6]. Infrastructure development is often hampered by stresses related to violence,

conflict, and instability [7]. Existing infrastructure is often obsolete, dismantled, and sold off, and, where functional, is hard to maintain as the staff are at risk. Power systems, in particular, face major challenges as their development is subject to high risk of attacks or failure [8]. Predicting the exact location of these failures is of a stochastic nature and is therefore difficult to spatially pinpoint and predict [8]. However, some areas are more sensitive to conflict than others and this information is inherently affected by location. Thus, it is possible to capture this information qualitatively and use it to support the electrification planning process. This, in turn, requires the existence of a spatially explicit energy system modelling framework that is able to integrate this information in its objective function.

In the past few years, the advent of geospatial data and information technology has stimulated the development of methodologies, techniques, and tools aimed at supporting geographic information system (GIS) based electrification planning (see Network Planner, Reference Electrification Model (REM), GEOSIM, and Open Source Spatial Electrification Tool (OnSSET)) [9]. These tools have enabled the creation of quantitative electrification plans with spatial specificity and accuracy. Their geo-spatial nature has, on the one hand, added a level of complexity in the modelling process and, on the other hand, has expanded the range and spatial diversity of input information, e.g., demographic attributes, poverty and wealth, topography, and access to infrastructure and resource availability, among others.

This paper introduces a methodological addition to an existing GIS based modelling framework. The Open Source Spatial Electrification Tool (OnSSET) was selected due to its open source nature and modularity, which facilitates experimentation with its code base. In particular, geospatial information about fragility is introduced as a new input parameter and used as a lever for risk-adjusted discounting rates and risk premiums applied in each location. Note that this paper is not oriented towards finding which technology can deal with conflict better; it is instead focusing on indicating how high-level electrification mix dynamics change if conflict-induced cost adjustments apply. Under this premise, the suggested additions are tested against the case study of Afghanistan, a country that remains extremely fragile and faces enormous development challenges. A scenario-based approach is adopted, aimed at providing insights to the following questions:

- What is the role (quantitatively and qualitatively) of on- and off-grid systems in the electrification of Afghanistan (SDG 7.1) and to what degree does conflict change this?
- Where should electrification policy focus in order to support stakeholder (public, private sector, and international aid) cooperation and promote investment for power infrastructure modernization in Afghanistan (SDG 7.B)?

Section 2 provides some background information about energy planning in fragile states and sets the rationale of this paper. Section 3 presents the methodological additions; first the main modifications are formulated and then they are tested on Herat and Helmand provinces for proof of concept. Section 4 presents results and findings after scaling up the analysis for the whole country. Finally, Section 5 analyses key research findings, discusses electrification policy mandates, highlights the study's limitations, and provides recommendations for future work.

## 2. Power System Planning in Fragile States

### 2.1. Background

Conflict prone areas rank quite low in terms of modern energy service delivery. Take, for example, Somalia, Afghanistan, South Sudan, and Yemen, where spiraling conflict has stalled development for years and has kept access to reliable electricity at very low levels, especially in rural settings. Figure 1 illustrates the relation between security fragility (expressed though the Country Policy and Institutional Assessment (CPIA), which measures to what extent the institutional framework a country adopts supports sustainable growth and poverty reduction [10]. Every year the Center of Conflict, Security, and Development (CCSD) at the World Bank elaborates the "Harmonized List of Fragile Situations", which includes all countries with a CPIA score below 3.200 [11]; lower CPIA value indicates higher

fragility) and access to grid electricity. The enclosed numbers illustrates non-state violent attacks to infrastructure since 1980.

Security fragility has also given rise to, or has emboldened, militia groups. They leverage instability in order to increase power and political influence. In some instances, this has been expressed through attacks on energy infrastructure. Such attacks (reported in the press) might be incited by crime (like piracy in Somalia [12] and Nigeria [13]), political messaging (Myanmar [14], Colombia [15]), or undermining government activities (Afghanistan, Yemen, Sudan, Thailand, Iraq, Syria, South Africa [16,17]).

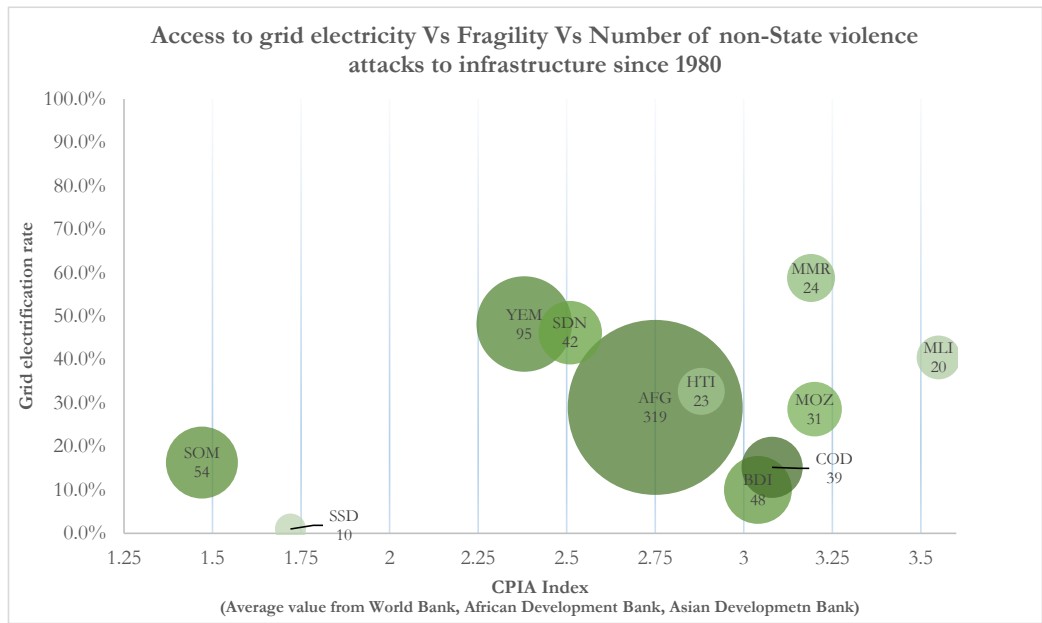

**Figure 1.** The figure illustrates the countries that have experienced more than 10 non-state (Instigated by non-state actors (state independent individuals or groups) to attain a political, economic, religious, or social goal through fear, coercion, or intimidation as per definition in [17]) violent attacks to facilities/infrastructure since 1980 (indicated by the size of the bubble). All indicate a low Country Policy and Institutional Assessment (CPIA) index and high grid electrification deficit [18,19]. The case of Afghanistan is particularly outstanding. Since 2010, ten attacks to power infrastructure have been recorded, targeting primarily power pylons in Jawzjan, Logar, Kabul, Beghlan, and Laghman provinces [11,17,20]. Country codes as per ISO-3166 Alpha-3: Afghanistan (AFG), Yemen (YEM), Somalia (SOM), Burundi (BDI), Sudan (SDN), Democratic Republic of the Congo (COD), Mozambique (MOZ), Myanmar (MMR), Haiti (HTI), Mali (MLI), and South Sudan (SSD).

Since 2010, an estimated 1546 non-state violent attacks on energy infrastructure (power, oil, and gas) have been recorded worldwide (North Africa and the Middle East: 546, South Asia: 500, South America: 182, South East Asia: 153, Sub-Saharan Africa: 111, Eastern Europe: 44, North America: 7, Western Europe: 3 [21]). About 47.1% of these targeted electric power networks. Thailand, Afghanistan, Iraq, and Yemen seem to bear the highest burden, accounting for more than 155 attacks [21]. The consequences have been dire in terms of fatalities, destroyed facilities, lost earnings, and reputational damage, which all discourage new investments in the power sector [21].

### 2.2. Rationale

Failures in power infrastructure lead to electricity supply interruptions, which, depending on their intensity and duration, can have devastating social and financial consequences on the affected area. The effects can be even worse in areas of high fragility. Interruptions may last from weeks to months or, in some cases, may stay permanently irreparable. In these cases, there might not be

adequate financial resources to secure restoration, or there might be reluctance by the investing body to restore failures, as the probability of another violent incident is high. It might be also the case that institutional inefficiencies and lack of coordination are exacerbated due to increased focus on security issues at the expense of others. Failure of power infrastructure may also affect the provision of other services such as water, health, and communications (ICT) [22,23]. Yet, timely restoration of electricity services is crucial to basic daily needs, business and economic recovery, and peoples' behavior and wellbeing [24].

Underestimating the importance of security (and thus resilient energy systems) can lead to inappropriate energy infrastructure deployment plans. Such plans are usually based on general economic indicators (such as lowering overall costs) [8,25]. Their construction, operation, and overall economic performance, however, is often vulnerable in unstable settings [8]. Take, for example, Afghanistan, where power sector plans recommend the development of power transmission networks through conflict prone areas [26]. The prospect of its commissioning in the short term is poor. Meanwhile, while waiting for grid electricity to arrive, populations in many areas turns towards different sources to cover their daily energy needs; about 60 percent of the population in Afghanistan now has some access to electricity through a solar device [19].

Unlike the traditional centralized grid, a partially decentralized power system architecture may be more suitable in fragile states [8,27]. A traditional grid system requires infrastructure to be 'interconnected'. This includes the extraction or import of fuel, its transport, the power plant used to convert it to electricity, and the extensive web of high (HV), medium (MV), and low voltage (LV) grid lines. The sabotage of any of these will result in power to the user being cut. For instance, transmission lines carrying electricity from Uzbekistan to Kabul cross through unstable areas (Dand-e-Ghori and Dand-e-Shahabuddin in Baghlan province) and are often targeted [28]. The cost of guarding and/or repairing power infrastructure there is high.

Decentralized power networks are, on the other hand, modular. They do not require an extensive web of grid lines (Although their long term sustainability does require well-sustained supply chains). Rather than one large power plant, they consist of many smaller power plants. Given the cost revolutions, especially for solar, today's fuel of choice can be readily renewable (solar, hydro), while in the past a lot of isolated systems were purely run on diesel. This limits (or eliminates) the dependency on imports or transport in the area. If they are damaged, their repair or replacement is independent of the rest of the system as they can be easily isolated. This can limit the extent of damage from physical sabotage (This argument drives the scenario assumptions in following sections. Note, however, that knowledge management, business model integrity, and operational performance may also carry important risks in conflict states. These are not considered in this paper but are worthy of similar attention when developing holistic strategies). Consequently, it can make restoring power faster and less capital intensive. In addition, decentralization of power generation also means distribution of invested capital. Since unexpected disruptions are location explicit, lower capital density leads to lower overall financial risk. Finally, decentralization of the power system may lead to its democratization, as argued by [29]. With increased decentralization, local communities have, in selected instances, become more active in power generation and its evolution. Such was the case of small scale hydro power development in China during the 20th century [30], Germany today [31], and others. This may increase the sense of ownership within the community/area served and reduce the risk of an unexpected interruption for other-than-technical reasons.

Literature points to energy security (including system stability, flexibility, adequacy, and robustness) in respect of the impact of power interruption on the various functions of society [32–38]. Various methods have been suggested, including: critical infrastructure protection techniques [22], risk management and decision analysis frameworks [39], simulation techniques (e.g., Monte Carlo) [40,41], real options models [42,43], probabilistic and stochastic models [44], assignation of risk premiums and additional costs as tailored inputs [8], among others. [45] has conducted a thorough review of risk-based methods for energy system planning. Some studies focus on the effect security issues might

have on electrification efforts in developing economies [46–54]. However, only a handful of them have looked into fragile states [8,27,55,56]. [27] discussed the effect that deliberate attacks had on electric power systems (Bosnian war in the 1990s) and how reliability advantages offered by decentralized power networks can be included in modelling efforts with the use of a multiple probability simulation model. [8] explored (and indicated using the case study of Sudan) how typical inputs of least-cost planning models (such as interest rates, capital, construction times, and damages) can be affected in fragile states. [55] used stochastic optimization to produce a near-term hedging strategy against conflict-induced risks in South Sudan. [56] proposed a framework that considers multiple effects of conflict on power system planning and explored how conflict conditions that vary over time can influence power sector investment plans. However, none of the studies above provide an explicit framework that integrates the geospatial aspect of conflict into power system planning. In addition, existing GIS based modelling frameworks and their applications (selected examples in [9,57–63]) have mainly focused on the technical and economic aspects of electrification planning, overlooking socio-political aspects such as instability and conflict [56,64]. These gaps led to the methodological approach described in the following section.

## 3. Methods

### 3.1. General Overview

This paper extends the functionality of the Open Source Spatial Electrification Tool (OnSSET). OnSSET is a geographic information system (GIS) based tool developed to transform qualitative government goals into tangible policy actions for electrification. It works as follows: Household electricity demand levels are estimated for every geospatial unit (settlement) of an area of interest. The size of the unit depends on the spatial resolution of the input data. In this study, each settlement area is equal to 1 km$^2$. The model uses spatial information together with a series of technical, economic, and social parameters, all described in Appendix A. These are combined to spatially identify the least-cost electrification option(s) between three alternative configurations; they include grid connection/extension, mini grids (solar Photovoltaic or PV herein, wind turbines, diesel gensets, small scale hydropower) and stand-alone systems (solar PV, diesel gensets) [58].

The decision as to which technological configuration is the best fit in each location is based on the lowest levelized cost of electricity (LCoE) that can be achieved. Then, the model (Note that OnSSET only partially includes the issue of service reliability (intermittency) in stand-alone and mini-grid systems as explicit storage related costs. This is a core limitation of the model that we did not explore in depth in this paper) calculates the capacity and investment requirements for the selected technologies to be deployed to electrify the population to the desired level. It should be noted that, while sometimes limited, the LCoE is a useful metric for comparing electrification options in developing markets [65]. The core drivers of LCoE are technology (investment, operational) and financing (equity, debt) costs, all of which can be significantly higher in a fragile setting [66]. With this in mind, this paper induces two modifications of the LCoE metric. The first suggests the use of risk-adjusted discount rates; the second suggests the allocation of risk premiums on the fixed and variable cost of the electrification technologies. The rationale is briefly described below.

### 3.2. Risk-Adjusted Discount Rate

Power system development is a capital-intensive process that requires the mobilization of significant financial resources over long periods. These might include both public and/or private investments. Therefore, it is important that least cost electrification models take into consideration limitations in project financing and yield "bankable" investment plans that will not compromise electrification goals. This will help identify realistic investment requirements (and technology options) that would otherwise be missed.

Typically, both public and private investment is necessary in the form of either equity or debt capital. Investor decisions are influenced by downside risks such as construction delays, loss of assets, and default in payment by the customer [42]. These reflect the likelihood of a negative event occurring, multiplied by its associated financial impact [67]. The risks are integrated into financial flows as the cost of equity and/or cost of debt. Conflict increases downside risks; therefore, projects in fragile areas are induced to higher costs of capital [68,69]. These values affect, in turn, the weighted average cost of capital (WACC) [66]. In this analysis, a risk-adjusted version of the WACC is introduced, as shown in Equation (1). This is then used as an indicative discount rate in the LCoE formula. The rationale is that high fragility is expressed through higher discount rates, which in turn skews LCoE in favor of less risky technologies.

$$WACC_{ra} = (W_e \times k_e) + (W_d \times \beta_f \times k_d \times (1 - T_c)) \tag{1}$$

where, $WACC_{ra}$ = Risk-adjusted weighted average cost of capital (after-tax); $W_e$ = Percentage of financing that is equity; $k_e$ = Cost of equity capital; $\beta_f$ = Beta multiplier for security (Note that beta is a measure of volatility on the costs of capital. It was adopted by the Capital Asset Pricing Model (CAPM) under the assumption that security is highly correlated with the expected return of an investment. The default value is 1, indicating that $WACC_{ra}$ is estimated according to the standard costs of capital applied to electrification projects in a country. Beta greater than 1 indicates higher volatility and leads to higher $WACC_{ra}$. It should be noted that the above formula estimates $WACC_{ra}$ at real rates (excluding inflation). $WACC_{ra}$ at real rates can be estimated with the use of the Fisher hypothesis formula [70]); $W_d$ = Percentage of financing that is debt; $k_d$ = Cost of debt capital (before tax); $T_c$ = Corporate tax rate.

Note that a significant part of the electricity access infrastructure development is more likely to rely on grant funding or low cost capital from international sources, which are unlikely to change drastically with regards to the country's exposure to the conflict; under this assumption, the security beta was only applied on cost of debt capital.

### 3.3. Risk Premium on Fixed and Variable Costs

In order to alleviate the effects of an unexpected event, power infrastructure in fragile areas needs to be safe-guarded by using additional or more expensive system components (underground transmission, increased facility security, and other countermeasures [71]). Similarly, conflict increases maintenance costs (system component failure, lack of supply equipment and materials, and the inability to provide much of the necessary specialized personnel, guards, etc.). Fuel costs or shortages increase with poor accessibility to or compromised transportation networks.

In order to include these additional costs in the electrification model, a stress adjustment factor (SAF) is introduced to reflect risk premiums needed to safeguard systems. The factor applies to capital, Operation and Maintenance (O&M herein), and fuel costs, respectively, for each type of technology and varies according to the fragility status of a location. Thus, the LCoE formula is modified as follows:

$$LCoE_{tech} = \frac{\sum_{t=1}^{n} \frac{(C_cSAF_{tech}*I_t)+(O_cSAF_{tech}*O\&M_t)+(F_cSAF_{tech}*F_t)}{(1+r)^t}}{\sum_{t=1}^{n} \frac{E_t}{(1+r)^t}} \tag{2}$$

where $I_t$ is the investment expenditure for a specific system in year $t$; $O\&M_t$ is the operation and maintenance cost; $F_t$ is the fuel expenditure; $E_t$ is the generated electricity; $r$ is the discount rate; and $n$ is the lifetime of the system. $C_cSAF_{tech}$ is the Capital cost Stress Adjustment Factor for each technology; $O_cSAF_{tech}$ is the Operation and Maintenance cost Stress Adjustment Factor for each technology; and $F_cSAF_{tech}$ is the Fuel cost Stress Adjustment Factor for each technology.

Note that for off-grid systems, SAFs are directly applied to the relative costs of each technology. For the grid, SAFs apply only to the development costs of Transmission and Distribution (T&D herein) lines since the power plants are considered to be remotely located and more easily policed (than an extensive grid network). This assumption may be adjusted according to the circumstances that apply.

### 3.4. Testing Assumptions

The methodological modifications are first tested against two provinces in Afghanistan, namely Herat and Helmand (Table 1) by analyzing a number of electrification scenarios. The scenarios explore how different technologies react under (a) different discount rates and (b) increasing fixed and variable costs per type of technology. All scenarios assume 100% electrification by 2030 and a target demand of ~4430 kWh/year for urban and peri-urban households and ~1365 kWh/ year for rural households.

**Table 1.** Profile of provinces used to test methodological modifications. Listed characteristics have been estimated based on geospatial processing. Input values and other modelling assumptions are presented in Appendix A.

| Herat Province | Helmand Province |
|---|---|
| • North-west Afghanistan | • South-western Afghanistan |
| • Population connected to the grid: ~40% | • Population connected to the grid: ~24% |
| • Expected population in 2030: 3.1 million | • Expected population in 2030 - 1.4 million |
| • Relatively high population density | • Low population density |
| • Relatively developed infrastructure | • Limited and obsolete infrastructure |
| • Low-Medium fragility | • High fragility |

### 3.4.1. Least Cost Electrification Mix under Various Discount Rates

In Herat (Figure 2), changing discount rates affect predominately the share of mini-grid PV systems, especially in lower rate ranges. By increasing the discount rate from 5% to 10%, we observe a drastic reduction in the share of mini-grid PV systems by about 18%. A plateau is observed between 10% and 20% and then again there is a mild reduction of about 8% between the 20% and 25% discount rate. The share of the grid does not seem to be affected significantly by changes of the discount rate; it's share is slightly increasing as the discount rate increases. It should also be noted that diesel generators become a prominent technology for a considerable amount of settlements at higher discount rates (>20%). This is due to the nature of the LCoE metric, in which higher discount rates tend to "favor" technologies with low capital and high variable costs.

Stand-alone PV systems also seem to be quite tolerant in discount rate fluctuations; changes are observed only at very low and very high rates. At low rates, an interplay is observed between stand-alone and mini-grid PV systems because of the different lifetimes of these technologies; at higher rates we observe some change between stand-alone PV and diesel generators due to the LCoE metric functionality, as explained above.

A similar pattern is observed in Helmand (Figure 3). In this case, however, there is a more clear inflection point at a discount rate of 10%. Note that there is a linear relation between the share of mini-grid PV and the discount factor, which is inversely proportional; this leads to a strong reduction of mini-grid PV share, moving from 5% to 10% discount rate, where inclination is smoother. Proportionally, a higher increase of both grid and stand-alone PV systems is observed at higher discount rates, but there is no (or very little) influence on the prevalence of diesel generators.

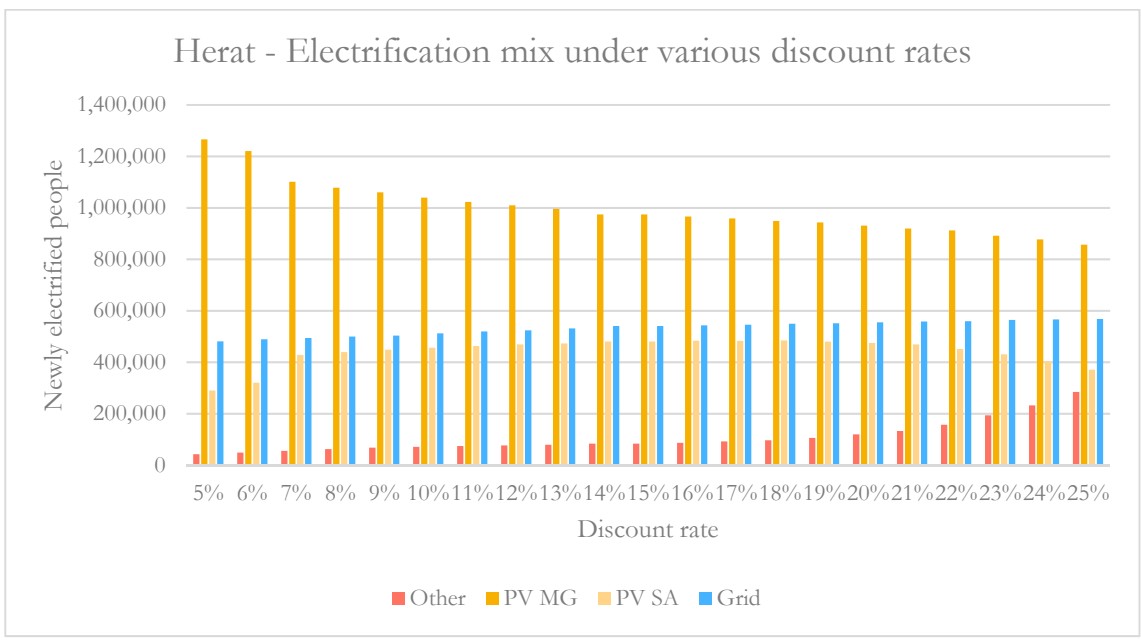

**Figure 2.** Electrification technology mix for Herat province in Afghanistan under the application of discount rates varying from 5% to 25%. "Grid" stands for grid network extension, "PV MG" for photovoltaic mini-grid systems, "PV SA" for photovoltaic stand-alone systems. The "Other" category includes primarily stand-alone diesel generators (~67%) and wind mini-grid systems (~33%); hydro and diesel based mini-grids were also included, although they were negligible (<0.008%).

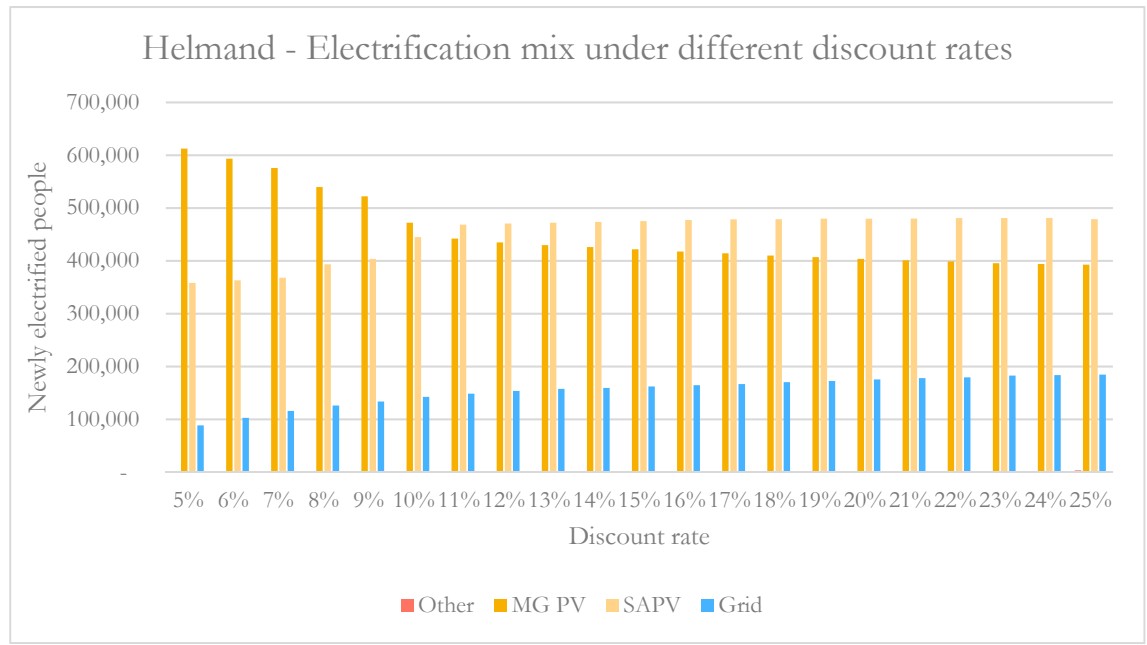

**Figure 3.** Electrification technology mix for Helmand province in Afghanistan under the application of discount rates varying from 5% to 25%. "Grid" stands for grid network extension, "PV MG" for photovoltaic mini-grid systems, "PV SA" for photovoltaic stand-alone systems. The "Other" category includes solely stand-alone diesel generators and wind mini-grid systems; hydro and diesel based mini-grids were also included, although they were negligible.

These results indicate that if fragility is expressed through discount rate increase (higher fragility, higher risk, higher discount rate), fewer mini-grid PV systems will be identified in the least cost

electrification mix. On the contrary, as fragility increases, stand-alone PV or diesel generators will be more prevalent. The latter depends on the geography of the studied area. For example, in Herat, diesel transportation costs are low due to a more developed infrastructure, which leads to more diesel gensets in the electrification mix as the discount rate increases. On the contrary, in Helmand, where infrastructure is limited and settlements are remote and sparsely populated, the share of diesel in the mix is very low. In addition, grid extension shows higher tolerance to increasing discount rates. That is, despite the increase of discount rates (which consequently pushes the grid LCoE value higher), connection to the grid still remains the least-cost option for many areas in our case studies; in fact, it even absorbs part of the share of lost PV mini-grid systems. The findings, of course, rely on the assumption that the electrification process is open to private investment in Afghanistan. However, it is more likely that a significant part of the country's electrification will rely on public funding or international aid. In this case, discount rates will most probably remain unchanged regardless of a location's exposure to risk. Therefore, in such cases, the introduction of risk premiums is, perhaps, a more sensible approach to examine policy implications related to optimal mix, capacity, and investment requirements.

### 3.4.2. Least Cost Electrification Mix under Risk Premiums

In the first set of scenarios, and in alignment with the narrative in Section 2, risk premiums are introduced only to grid extension; that is, capital and maintenance costs of T&D networks. Note that the discount rate in all scenarios that follow is set at 12% (Appendix A).

Figure 4 reveals a very small variation in the least cost mix in Herat. Assuming that 0% reflects baseline costs, a 10% increase will cause a 6.5% reduction in the grid's share. The shift favors mini-grid PV systems. After that, changes are marginal. The reason for this lies in the fact that about 20% of the unserved population in Herat lives in already grid connected areas or very close to the transmission grid; due to high target demand and short distance to existing lines, grid in these areas is the least cost electrification option even if risk premiums apply. Stand-alone PV systems and other technologies do not seem to benefit much from the addition of risk premiums on grid.

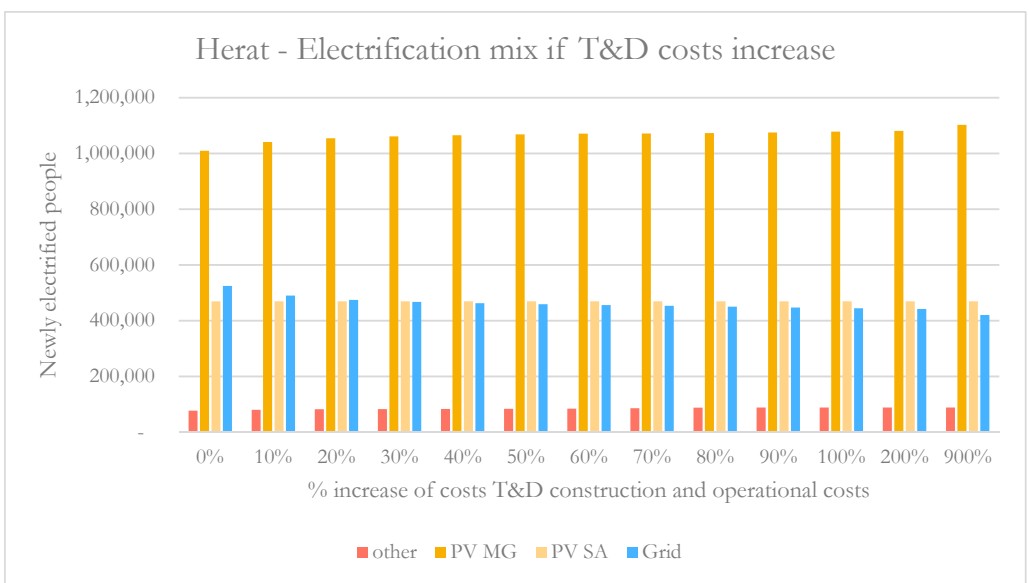

**Figure 4.** Electrification technology mix for Herat province in Afghanistan under the application of risk premiums to T&D construction and operational costs. The first bar (left) indicates results without risk premiums, values set at default (Appendix A). The rest of the columns indicate percentage increase over default values. "Grid" stands for grid network extension, "PV MG" for photovoltaic mini-grid systems, "PV SA" for photovoltaic stand-alone systems. The "Other" category primarily includes wind mini-grid systems; hydro and diesel based mini-grids and stand-alone systems were also included, although they were negligible (<0.008%).

In Helmand (Figure 5), however, if T&D costs increase by 20–30%, then the share of the grid in the electrification mix halves (from 14% to 6.5%). The shift favors mainly mini-grid PV systems. In this case, only about 6% of the unserved population lives in areas that are already electrified, or very close to those; the rest reside at variable distances at which the difference between grid extension and mini-grid PV is marginal. Hence, least cost options change. Similar to the previous case, no particular changes in stand-alone PV and other technologies are observed.

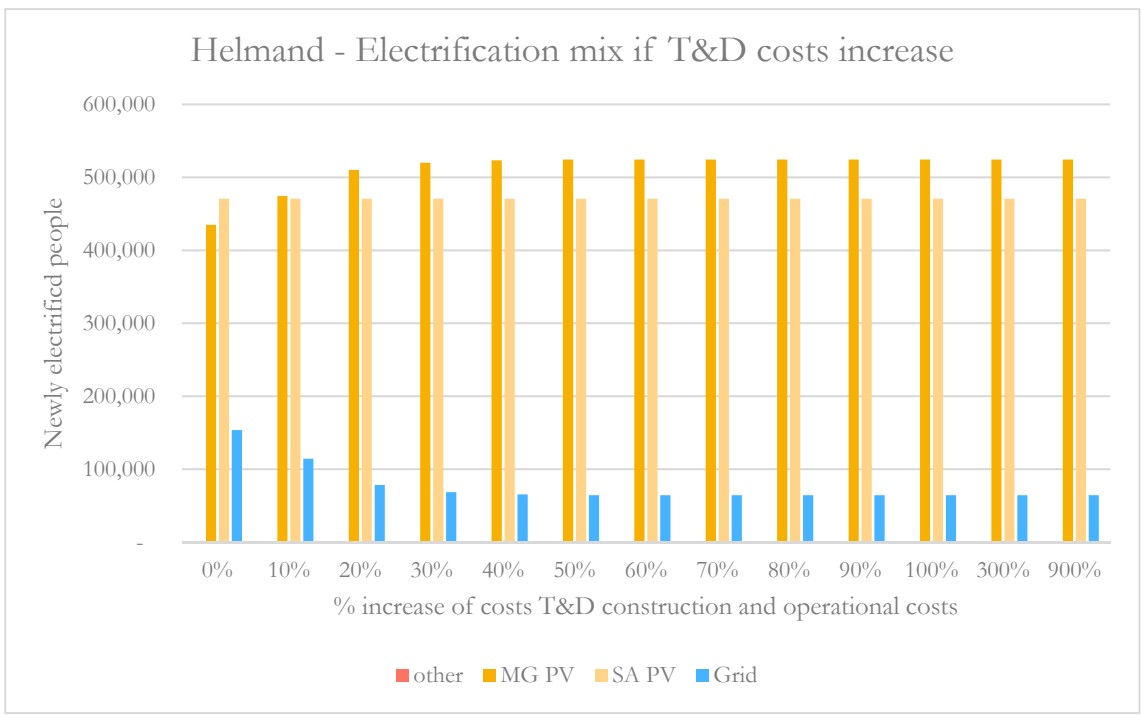

**Figure 5.** Electrification technology mix for Helmand province in Afghanistan under the application of risk premiums to T&D construction and operational costs. The first bar (left) indicated results without risk premiums, values set at default (Appendix A). The rest of the columns indicate percentage increase over default values. "Grid" stands for grid network extension, "PV MG" for photovoltaic mini-grid systems, "PV SA" for photovoltaic stand-alone systems. The "Other" category includes diesel generators and wind and hydro mini-grid systems, which are negligible in this case.

The results above indicate that the effect of risk premiums on the grid is expected to favor predominantly mini-grid PV systems. The magnitude depends, to a great extent, on the geography of settlements under consideration. As seen in the two examples, the same risk premiums can cause a different shift in least cost options for electrification e.g., 6.5% in Herat and 55% in Helmand.

A question arises, however; should risk premiums be applied only to grid or should they be introduced to off-grid systems as well? This question is examined in a second set of scenarios in which risk premiums are introduced to grid and mini-grid costs as well as to diesel fuel cost. No risk premiums apply to capital and maintenance costs of stand-alone systems. Note that this is a core assumption of this paper. It is based on the rationale presented in Section 2, where the literature suggests that big, capital-intensive power projects are more prone to physical sabotage. This is less likely to be the case for stand-alone systems due to their smaller size and disperse distribution. In fact, several such systems have already been deployed in rural Afghanistan, while waiting for the grid to arrive. This, of course, does not exclude stand-alone systems from being susceptible to risk due to conflict. However, in this paper, the lower probability of such systems being affected by conflict is represented by zero risk premium additions.

As expected, stand-alone PV systems dominate the share as risk premiums increase for the rest of technologies. What is interesting is the path under which the technology swap is happening.

Mini-grid PV systems show a steady linear decrease until the risk premiums reach about 40% and 30% in Herat (Figure 6) and Helmand (Figure 7), respectively. Then, PV mini-grids are no longer a viable electrification option for any settlement. It is noticeable that other mini-grids or diesel gensets disappear from the mix with risk premium of even 20% (in Herat). The grid is always retained for reasons mentioned above (e.g., high demand, close proximity, or existing connection). Interestingly, we observe some shift from grid to stand-alone PV if the risk premium reach more than 40–50%.

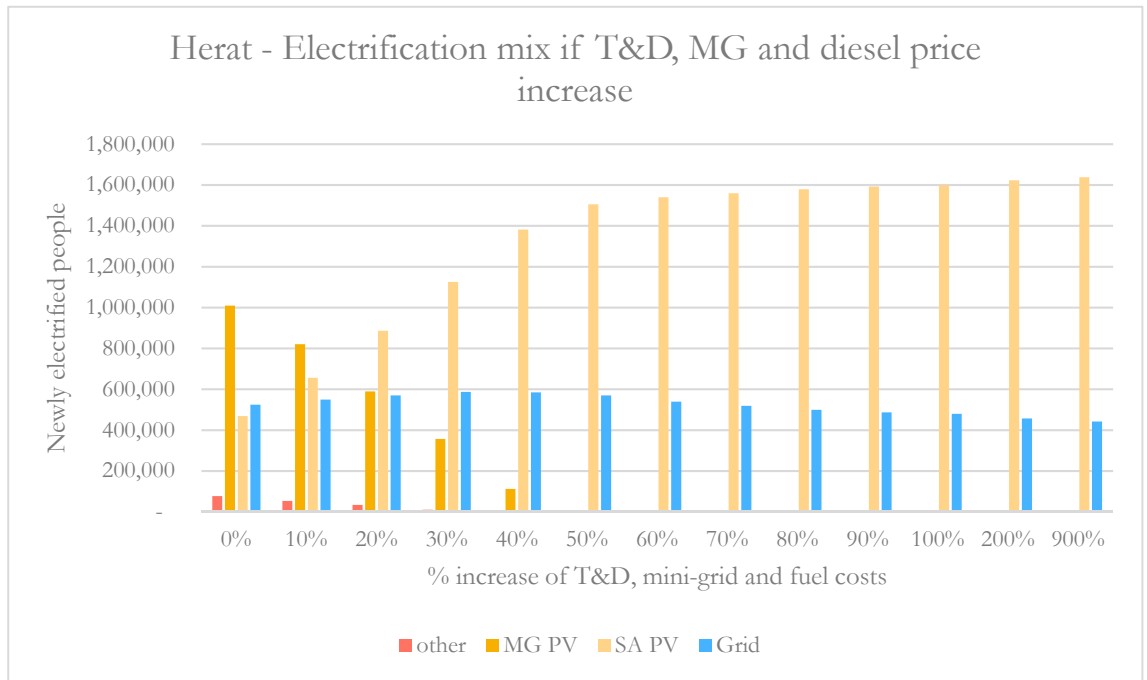

**Figure 6.** Electrification technology mix for Herat province in Afghanistan under the application of risk premiums to T&D, mini-grid, and fuel costs. The first bar (left) indicated results without risk premiums, values set at default (Appendix A). The rest of the columns indicate percentage increase over default values. "Grid" stands for grid network extension, "PV MG" for photovoltaic mini-grid systems, "PV SA" for photovoltaic stand-alone systems. The "Other" category includes primarily wind mini-grid systems; hydro, diesel based mini-grids, and stand-alone systems were also included, although they were negligible (<0.008%).

The two examples of Herat and Helmand indicate that the optimal electrification mix is very sensitive to the local context. At the same time, certain patterns emerge. Populations in urban areas create a strong consumer base for grid electricity even if reinforcement, densification, and extension is subject to additional "risk-mitigation" costs. Population in rural areas tends to be better served by stand-alone PV or small diesel gensets. However, a significant part of the population in-between (peri-urban areas) can be characterized as a "grey zone". There, incremental changes in costs or project financing schemes due to conflict can swiftly change the least cost option from one technology to the other; that, in turn, may require alternative funding sources and/or policy mandates.

In the following paragraphs, this experiment is scaled-up at a national level for Afghanistan and aims to identify the inflection points between these zones, to quantify key decision parameters, and to provide policy recommendations for electrification.

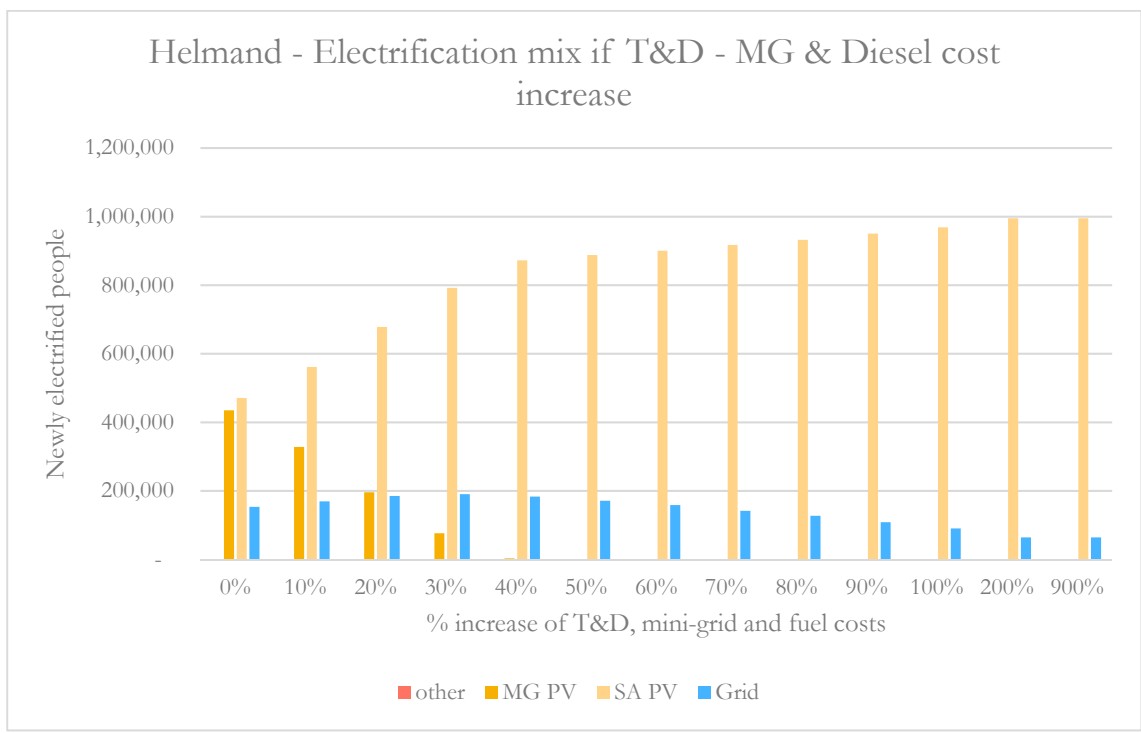

**Figure 7.** Electrification technology mix for Helmand province in Afghanistan under the application of risk premiums to T&D, mini-grid, and fuel costs. The first bar (left) indicated results without risk premiums, values set at default (Appendix A). The rest of the columns indicate percentage increase over default values. "Grid" stands for grid network extension, "PV MG" for photovoltaic mini-grid systems, "PV SA" for photovoltaic stand-alone systems. The "Other" category includes diesel generators and wind and hydro mini-grid systems, which are negligible in this case.

### 3.5. A Conflict-Adjusted Least Cost Electrification Plan for Afghanistan

#### 3.5.1. Background

Wracked by more than three decades of conflict, Afghanistan remains an extremely fragile state and faces enormous development challenges. Per capita electricity consumption varies significantly throughout the country; it averages at 497 kilowatt-hours (kWh) per year [26], which is lower than the South Asia average of 667 kWh per year and far below the global average of 3104 kWh [72]. Accessing the grid electricity remains a serious challenge in rural areas, where more than 73% of Afghans live. The rural population's share of grid electricity access is estimated at a mere 11% [19,57]. Even households with access to the grid electricity continue to suffer prolonged power outages, especially during the winter periods [73]. The last decade has given rise to an off-grid solar uptake, with small scale PV systems becoming available in many parts of the country (providing power for lighting and the ubiquitous mobile phone) [19,74]. Further advancements in rural electrification include the promotion of community-level micro hydropower [74]. Despite progress, it is still questionable whether these projects had a significant effect on achieving the full objective of SDG7 [75]. In addition, lack of any geospatial information around those makes it very difficult to locate them and record their performance. Therefore, for the purpose of this study, off-grid population is considered currently as "un-served".

#### 3.5.2. Scenarios

The baseline scenario assumes that no conflict restrictions apply. It represents a standard approach to geospatial electrification planning [60,61,76], assuming peace over the modelling period. Parameters were set at default values, as presented in Appendix A.

Scenario A introduces risk-adjusted discount rates based on the level of fragility in each location. The default value of 12% was gradually increased based on the level of conflict in each location (see Figure A3, Appendix A). The debt equity ratio and cost of equity capital were assumed to be constant. [77] indicates that the cost of debt can vary by around 15% between the low to higher risk projects. Therefore, a gradual increase of 15% in the costs of debt capital per fragility level were introduced, as seen in Table 2. $WACC_{ra}$ was then calculated for each zone and used as the discount rate in the LCoE formula. All technologies were subjected to the same $WACC_{ra}$ in a given location.

**Table 2.** Indicative example incremental increase in project financing parameterization. Note that $WACC_{ra}$ is the risk-adjusted weighted average cost of capital as defined in Section 3.2.

| Fragility Index | Cost of Equity Capital | Beta for Security ($\beta_f$) | Cost of Debt Capital | $WACC_{ra}$ (Discount Rate) |
|---|---|---|---|---|
| | All Technologies | All Technologies | All Technologies | All Technologies |
| Total unrest | 12.0% | 1.60 | 24.0% | 17.1% |
| High fragility | 12.0% | 1.45 | 21.8% | 15.8% |
| Medium fragility | 12.0% | 1.30 | 19.5% | 14.5% |
| Low fragility | 12.0% | 1.15 | 17.3% | 13.3% |
| Neutral fragility | 12.0% | 1.00 | 15.0% | 12.0% |

Similarly, Scenario B introduces risk premiums on electrification technologies based on the level of fragility in each location. It was assumed that power infrastructure in low, medium, and high fragility areas in Afghanistan is subjected to risk premiums that range between 18% and 60% (rationale in Appendix A). The scenario was built by additionally introducing two extreme cases; the total unrest case where risk premiums reach 100% and neutral fragility areas where no risk premiums are assigned. Table 3 provides a summary of the cost component adjustments suggestions.

**Table 3.** Indicative example of stress adjustment factor (SAF) values for capital, operational, and fuel costs of the three electrification configurations used in the Open Source Spatial Electrification Tool (OnSSET).

| Fragility Index | Risk Premium (% of Initial Value) | | | | | |
|---|---|---|---|---|---|---|
| | Grid and Mini-Grids | | | Stand-Alone Systems | | |
| | Capital Cost SAF | O&M Cost SAF | Fuel Cost SAF | Capital Costs SAF | O&M Costs SAF | Fuel Costs SAF |
| Total unrest | +100% | +100% | +100% | +0% | +0% | +100% |
| High fragility | +60% | +60% | +60% | +0% | +0% | +60% |
| Medium fragility | +39% | +39% | +39% | +0% | +0% | +39% |
| Low fragility | +18% | +18% | +18% | +0% | +0% | +18% |
| Neutral fragility | +0% | +0% | +0% | +0% | +0% | +0% |

Note that grid extension activities and mini-grid systems follow the same risk premium scheme because the deployment of both is affected by fragility. For the grid, premiums safeguard infrastructure robustness and resilience; for mini-grids, premiums safeguard self-sufficiency (through higher storage capacity) after the occurrence of sudden disturbance [78]. By contrast, it was assumed that stand-alone systems are not subject to any risk premiums in terms of capital and operational costs. However, fuel costs (in this case, diesel cost) were assumed to be subject to the risk premium.

Finally, Scenario C assumes that conflict is expressed through both discount rate and risk premium adjustments.

## 4. Results

Nationwide results are in alignment with the findings in Section 3. That is, across all scenarios, grid connection seems to be the most cost effective electrification option for a large part of the Afghan population, especially around the main urban centers (Kabul, Kandahar, Herat, Mazar-i-Sharif, and Kunduz). However, the grid is far from sufficient to achieve the country's electrification goals.

Photovoltaic systems (both mini-grid and stand-alone) have also been identified as least cost options in several locations throughout the country (e.g., Kandahar, Helmand, Khost, Paktya, Paktika, and Logar, to name a few). Mini-grid wind systems are the least cost option in some locations, with high wind resource availability mainly in Farah and Sari Pul. Similarly, mini-grid hydro systems seem to provide the least cost electrification option, mainly in northeastern Afghanistan, in mountainous areas with high resource (water) availability (e.g., Badakhshan, Takhar, Nuristan, Kunar, and Panjshir). Off-grid diesel generators can only be competitive in few sparsely distributed locations around the country. Summaries of electrification results per province are available in Appendix B.

*4.1. Least Cost Electrification Mix*

According to the baseline scenario, 50.9% of the total Afghan population in 2030 can be electrified by the grid. Photovoltaic systems can electrify 39.5% of the population in the form of mini-grids and 8.9% in the form of stand-alone systems. The contribution of other mini-grids is small; wind based mini-grids can contribute to about 0.6% of the total new connections, while hydro based mini-grids can contribute to 0.04%. This is due to the fact that wind and hydro systems are only LCoE competitive in areas with high resource availability in comparison to solar irradiation, which is available anywhere. Only a few diesel stand-alone systems were identified as an economic solution and no diesel mini-grids; this is due to the high diesel price set in the model and despite the fact that the discount rate was set at 12%.

According to fragility status (Figure A3, Appendix A), approximately 46.1% of settlements in Afghanistan are affected by conflict. This explains the results in Scenario A, which indicated a few minor, yet noteworthy, differences to the baseline. In particular, according to this scenario, grid extension is expected to electrify 51.7% of the population in 2030. PV systems play an important role in this case as well, with their deployment reaching 38.3% of the population as mini-grids and 9.1% as stand-alone systems. The share of wind power mini-grids is slightly higher than the baseline scenario at 0.7%. An increase is also observed for stand-alone diesel systems, which are expected to electrify 0.1% of population. The share for hydro remained low; no diesel mini-grids were identified.

The changing discount rates seem to have the highest impact on the share of mini-grid PV systems in the total electrification mix. In particular, about 10,317 settlements that identified mini-grid PV systems as the most cost-effective option in the baseline scenario changed towards other technologies in Scenario A. As shown in Figure 8, the percentage change of the average LCoE values in the different conflict regions seems to be (in most cases) higher for mini-grid PV systems than other competitive technologies. Take, for example, areas under total unrest; the higher discount rate introduced in these regions caused the average LCoE value of mini-grid systems to increase by 22.5%, higher than any other technology except hydro. This consequently reduced the share of mini-grid PV systems by 6.4% (affecting approx. 61,700 people), with the optimal solution swapping mainly towards grid extension and stand-alone diesel generators. Similar patterns were recorded in the other regions as well. It should be noted that the grid extension is also affected by the "cascading" effect. That is, a change in few cells from mini-grid to grid extension may affect the least cost electrification option in neighboring cells due to the nature of the electrification algorithm used in the model. This can occur in any area, regardless of the fragility index. It should also be mentioned that diesel based systems show the lowest percentage change of average LCoE values. However, despite that, their overall share remains quite low in all cases because their generating costs are often higher than other technologies (Figure 8), mainly due to the high diesel price assumed in the model.

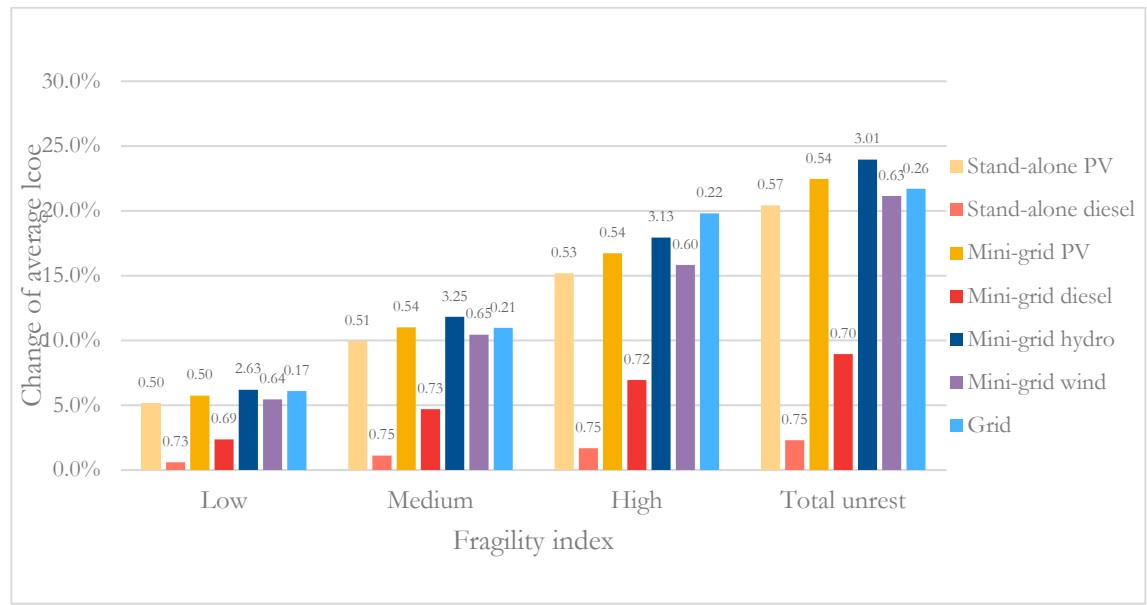

**Figure 8.** Percentage change of average levelized cost of electricity (LCoE) for all electrification technologies per conflict region. The label values above the bar indicate the average LCoE of each technology in each area in Scenario A. The percentage rate on the y-axis indicates how the average LCoE value per technology changes in comparison to the baseline scenario.

In Scenario B, 53.2% of the total population in 2030 can get electrified by the grid. PV mini-grids constitute 24.2% and PV stand-alone constitute 22.1% of the total connections. Other systems are expected to electrify less than 0.5% of the population in this scenario. These results indicate that mini-grids are more sensitive to the introduction of risk premiums than the grid, especially in areas with a higher fragility index (as shown in Figure 9).

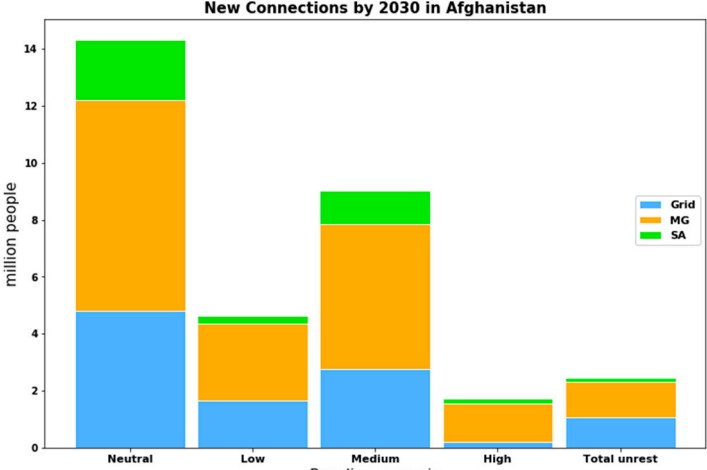

**Figure 9.** *Cont*.

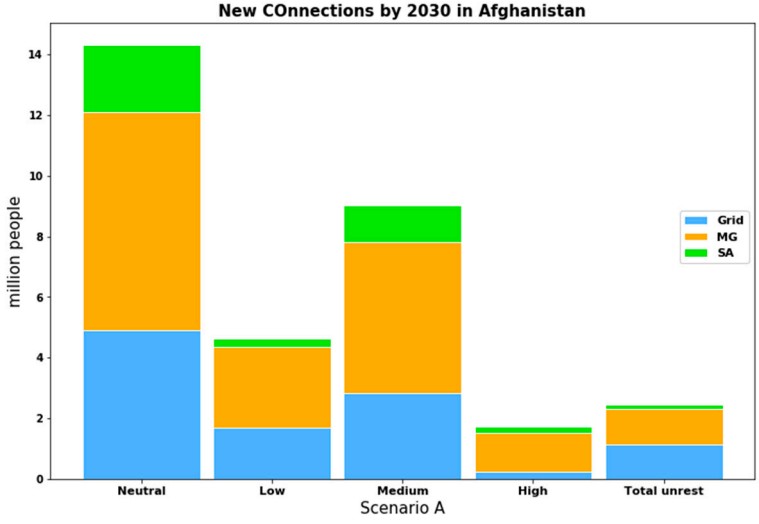

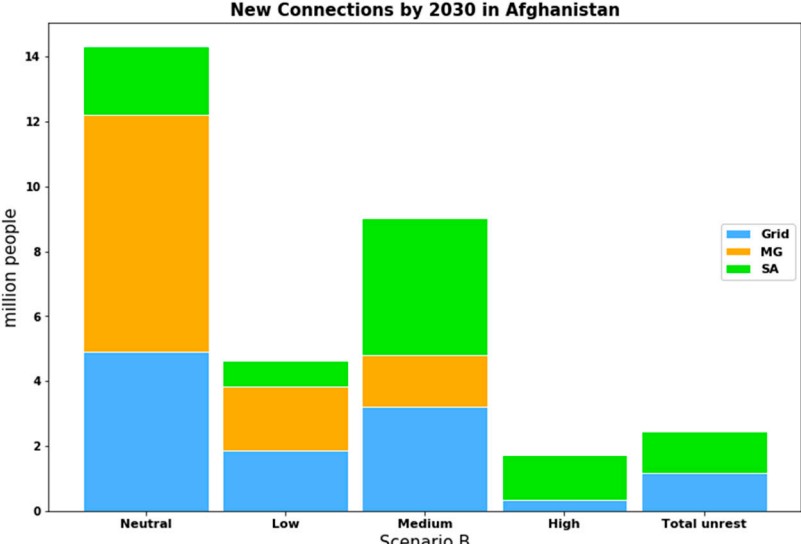

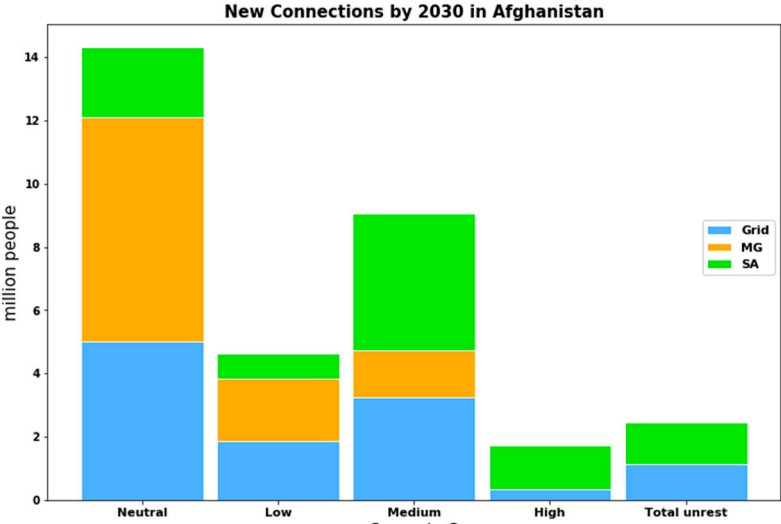

**Figure 9.** Least cost technology mix for the achievement of universal electrification in Afghanistan based on the four scenarios examined in this paper.

Finally, in Scenario C, grid connection can electrify 53.5% of the total population in 2030, mini-grid PV 23.4%, and stand-alone PV 22.1% of the total new connections. The share of wind power and hydro mini-grids is 0.5% and 0.3%, respectively. Stand-alone diesel is expected to cover 0.5% of the population. Spatial visualization of the least cost electrification option for all scenarios is available in Appendix C.

### 4.2. Capacity and Investment Needs

The total investment required to achieve full electrification under the baseline scenario is 14.6 billion USD. The investment necessary to connect 10.5 (newly electrified) million people to the grid by 2030 is approximately 4.4 billion USD. In parallel, decentralized technologies will require 10.2 billion USD to electrify 21.8 million people by 2030. Significant investment (~7.8 billion USD) shall be allocated for the deployment of mini-grid PV systems, with an estimated 1930.6 MW of capacity expected to be added. Stand-alone PV systems are expected to add 356 MW in the country's generating capacity, demanding approximately 2.32 billion USD by 2030. Finally, about 112.5 million USD shall be dedicated to the deployment of 24 MW and 0.6 MW of wind and hydro mini-grids, respectively.

In Scenario A, the total investment is reduced by 73.3 million USD in comparison to the baseline scenario (Figure 10). This is explained by the fact that part of PV systems (both stand-alone and mini-grids) were replaced by less capital intensive competitive technologies; mainly stand-alone diesel generators and grid densification. The difference is also reflected by the total new capacity needed, which dropped from 3553.0 MW in the baseline scenario to 3533.6 in Scenario A.

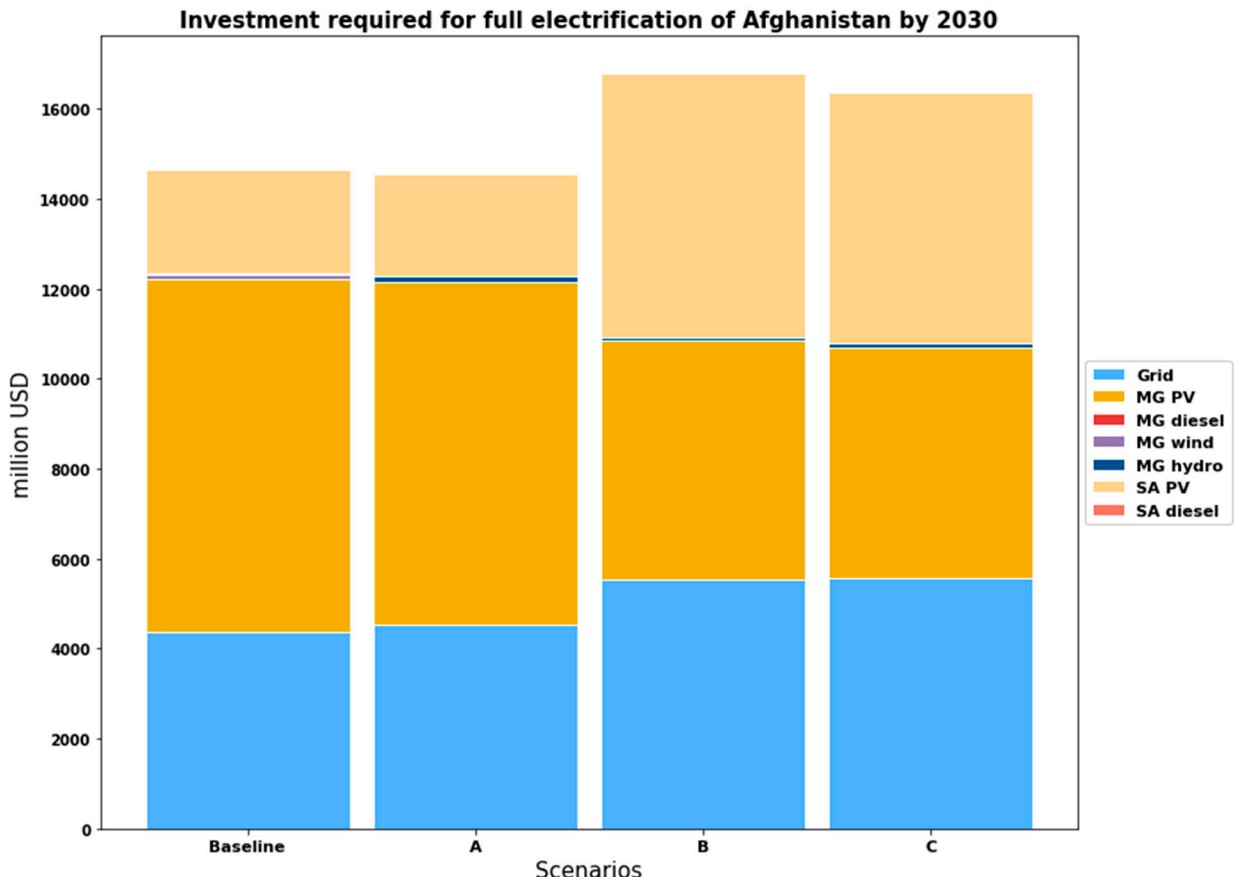

**Figure 10.** Total investment required for the full electrification of Afghanistan by 2030 under the four scenarios studied.

In Scenario B, as expected, the risk premiums pushed the total investment to 16.8 billion USD Figure 10). Investment for grid extension increased to 5.5 billion USD whereas investment for off-grid technologies increased to 11.3 billion USD. Significantly increased is the investment need for stand-alone PV systems by 3.6 billion USD in comparison to the baseline scenario. The total new capacity added in this scenario was estimated at 3445.7 MW.

Finally, in Scenario C, the total investment was reduced by 377.4 million USD in comparison to Scenario B for similar reasons related to the introduction of varying discount rates, as explained above. The total new capacity added in Scenario C was estimated at 329.1 MW.

Finally, noteworthy is also the role of renewable based technologies in the achievement of electrification targets in Afghanistan. In the case of grid, about 71.2% of indigenous capacity in 2030 is renewable based (forecast based on current government plans [57]); in the case of decentralized generation, in the scenarios studied, at least 99.7% of the added capacity comes from renewable sources. This is due to the high diesel price used in the model, which led to the adoption of very small diesel generators in about 5829 remote locations in northeastern Afghanistan. This might provide the basis for additional climate-mitigating concessionary financing schemes in Afghanistan.

## 5. Analysis and Discussion

The modelling exercise suggests that densification and extension of the grid network in Afghanistan is necessary as it can electrify more than 50% of the newly electrified Afghan population by 2030. For settlements in close proximity to the grid (less than ~50 km), grid extension is usually the least-cost option even when conflict risk is higher. Nevertheless, a large share of the population can potentially be electrified via off-grid systems (either mini-grids or stand-alone). The relative share of mini-grids and stand-alone systems depends considerably on the local context. The analysis shows that, where higher fragility prevails, a higher share of stand-alone systems is economically preferred. This conclusion relies on the assumption that additional fragility costs, as expressed in this study, have a greater impact on mini-grids than stand-alone systems. Another key parameter is the targeted level of electricity consumption. As indicated in [57], lower access targets lead to a higher share of stand-alone systems in Afghanistan. This highlights the importance of a systematic plan (with clear, feasible targets and timetables) that will enable the deployment of both on- and off-grid technologies in an institutionally, technically, and economically sustainable manner.

### 5.1. Towards a Policy for Electrifying Afghanistan

The following paragraphs present the author's interpretation of the results in regards to policy mandates for the main key stakeholders in the energy sector of Afghanistan.

First, results indicate that the public sector should focus on safeguarding the grid infrastructure. From a policy perspective, this requires:

- The development of a risk-inclusive, cost-benefit, prioritization plan for grid densification along with realistic targets and timetables for the full electrification of settlements that are already connected to the grid and grid extension. To avoid excessively optimistic expectations (and subsequent disappointment), it is important that all energy sector stakeholders know who, where, and when the public utility can electrify.
- Set tariffs that reflect the real cost of electrification and include risk premiums and/or cross-subsidization. The model indicates that densification costs an average of 267.6 USD/capita, while new grid connections require, on average, 481.6 USD/capita (for those within a 50 km radius of the grid). Note that a higher fragility index induces higher grid connection costs that can reach 514.4 and 615.0 USD/capita in areas with high fragility or total unrest, respectively.

Second, restructuring the power sector in alignment with electrification mandates requires the active participation of the private sector as well [73], particularly for the deployment of decentralized technologies (mainly mini-grids or distribution concessions for dedicated geographical areas). The

results of our analysis indicate that private investment should focus on less fragile areas first, where mini-grids (mainly PV) seem to be by far the least cost electrification option. Then, consciously consider areas with low or medium fragility, where mini-grids can still be the least cost option even if light risk premiums apply. Areas with high fragility/total unrest seem to be a no-go for private investment. In order to support private initiative, the electrification policy in Afghanistan should consider the following:

- Develop explicit regulation regarding licenses and permits for mini-grid operation. This calls for clear distinction in mandates, jurisdiction, and interactions between all involved stakeholders (e.g., the Ministry of Rural Rehabilitation and Development (MRRD) has a mandate for provision of services (including energy) to rural populations, which overlaps with the Ministry of Energy and Water (MEW) and Da Afghanistan Breshna Sherkat (DABS) mandates). To achieve better results for off-grid electrification, the establishment of a dedicated rural electrification agency, reporting to MRRD and MEW, respectively, should be envisaged.

- Results of the analysis indicate that average LCoE can increase considerably with higher fragility (Figure 8). Thus, policy should design (or allow for) flexible tariff schemes that are risk reflective (e.g., off-FiT [79]). Dedicated partial subsidies in fragile areas may be considered to reduce the comparatively higher cost.

- Implement strict technical, safety, and quality standards for mini-grid licensing and operation.

- Provide concessions (import tax reduction, capital subsidization, revenue-based financing) and/or compensation in the event of terrorism (repair grants).

- Develop clear guidelines on grid interconnection (e.g., distributor, generator, buyout model, or mix), because most of the prospective private mini-grid investments are expected in peri-urban areas.

Third, international aid is also needed to scale up electrification in Afghanistan in the suggested timeline [73]. Electrification policy in this case should

- Direct donor support in rural areas or areas with higher fragility that private investors or public utility cannot support immediately. This can be in the form of subsidies (e.g., up-front payment of grants) for smaller scale off-grid systems. To illustrate, mini-grid PV systems require on average of 481.5 USD/capita, while wind and hydro mini-grids require 455.6 and 424.2, respectively. Stand-alone PV requires, on average, 565.1 USD/capita, four time more than stand-alone diesel with about 129.9 USD/capita.

- Mandate donor's involvement in efforts that secure long-term sustainability of such systems (e.g., capacity building and maintenance) as well as scale-up over time as demand progressively grows.

*5.2. Final Remarks & Conclusion*

Achieving universal access to modern, reliable, and affordable electricity services is a challenging task. It requires the mobilization of significant financial resources and the coordination of multiple stakeholders. The latter can be particularly demanding in countries that face high political and social instability, such as Afghanistan. This paper introduced modifications to an existing modelling framework and an illustrative example of how the geospatial aspect of fragility might affect electrification planning in unstable states. By assigning risk-adjusted discount rates and premiums by technology and location, we have shown that urban settlements in Afghanistan create a stronger consumer base for on-grid electricity, even under higher risk. Peri-urban and rural areas are more sensitive to conflict-induced risk, therefore, off-grid electricity may be a more suitable solution for electrification within the next decade or so. This finding resonates with the premise that rural, less developed, and fragile areas are less likely to attract big, long-term investments that centralized grid networks require. Our approach is, however, not exhaustive; input data, methodology, and results are bound to limitations, a few of which are presented below.

In terms of input geospatial data, as in any analysis based on GIS, uncertainty is an integral component of the results. For the purposes of this paper, and together with Afghan stakeholders, the

authors have collected and processed the best datasets available, aiming to represent the status of the power sector in Afghanistan as accurately as possible, given data constraints. However, geo-spatial inaccuracies and data gaps cannot be entirely avoided; this should be taken into consideration when interpreting the electrification results. In addition, future work might deploy new and more efficient techniques for data acquisition, cleaning, and processing (e.g., advanced remote sensing) as well as quantifying and reducing uncertainty related to the geospatial aspect of this work.

In order to compensate for its high spatial granularity, the selected model (OnSSET) relies on a fairly simple "objective" function. That is, it identifies the least cost electrification approach only by comparing the LCoE values per technological options available. Selecting this model implies that its limitations are inherited as well. For example, conflict (or risk) in this case could only be integrated in the decision algorithm as a one-dimensional monetary penalty measure (see Section 2). On the one hand, this approach allows for a high level of input customization; the modeler can quantitatively determine risk parameters (in this case, discount factor and premium costs) per location and type of technology. On the other hand, this may lead to curtailed insight, especially in regards to previous work [25] exploring dynamic trade-offs between political uncertainty and cost minimization through multi-criteria optimization models.

Further, there is the issue of objectiveness in our approach. This raises questions similar to: How can qualitative information of conflict be appropriated into a scalar index? What determines the relation between this scalar index and discount rate or cost adjustments? What is their value range and how does this change per technology? How can one model different risk response for the technologies involved? The answer to these questions is not obvious. In this paper, we have looked at other examples (see Appendix A.3) to create a benchmark for power sector cost variation in Afghanistan. These values can only be indicative. However, considering information constraints, they were used to create a contextual framework for the case study in focus. Similarly, the allocation of risk premiums per technology was bound to the authors' subjective view. For example, risk premiums were only applied to grid and mini-grid configurations (to the same degree), when conflict can arguably affect stand-alone systems as well, but in the form of un-sustained value chains. That said, we understand that parameterization of such a model is likely to depend on the experts' judgments foremost. This calls for the development of a consistent evidence based risk-premium weighting as future research. Together with the proposed planning model, this could, in future work, take into account implementation realities on the ground, i.e., the issues with which planners in developing countries are faced. That is, depending on the context, de facto conflict risk premiums may, in some cases, be higher for (private) off-grid configurations compared to (public) on-grid solutions. For this reason, and in order to support, review, update, and/or reproduce similar research, along with this publication, both the input data [80] and the model's code base [81] are being made publicly available and open. Using those, a scenario discovery approach would be a very useful and sensible next step so as to highlight critical parameterization aspects of the model. Future work might also include sensitivities based on empirics of power sector planning in Afghanistan or other states facing similar fragility constraints. Lastly, aspects of this methodology could also be used in order to incorporate other potential sources of risk (e.g., climate vulnerability) in the geospatial electrification model.

Furthermore, it should be highlighted that the issue of service reliability (or the value of lost load) was only partially included in our model, through additional cost for storage in off-grid technologies. This is a limitation of the selected modelling framework (OnSSET) that we did not explore in depth in this paper. Yet, it is particularly important in the development of sustainable electrification plans, especially in such states. Furthermore, aspects related to probability and/or evolution of conflict over time were not covered. Future analysis may want to explore the combinations of remote sensing and machine learning techniques to provide spatial- or spatio-temporal predictions of conflict.

Despite its limitations, we believe that the suggested methodology takes geospatial electrification modelling a step ahead and helps build more information-inclusive strategies towards the achievement of SDG 7 in countries that need it the most.

**Author Contributions:** Conceptualization, A.K. and M.H.; Methodology, A.K.; Software, A.K. and D.M; Validation, A.K., D.M., M.H., M.B., A.S., S.F.H. and F.M.-R.; Formal Analysis, A.K.; Investigation, A.K.; Resources, A.K., D.M., M.B., A.S. and S.F.H.; Data Curation, A.K.; Writing—Original Draft Preparation, A.K. and M.H.; Writing—Review & Editing, A.K., D.M., M.H., M.B., A.S., S.F.H. and F.M.-R.; Visualization, A.K.; Supervision, M.H. and M.B.; Project Administration, M.H.; Funding Acquisition, M.H., D.M., M.B. and F.M.-R. All authors have read and agreed to the published version of the manuscript.

**Funding:** This paper was enriched by findings deriving from research activity funded by the World Bank under the contract number 7180875; the output of the research activity is openly available at [57]. Liaison with Afghan analysts was supported by DFID/UKAID under the project "Energy and Economic Growth (A0534A); National Energy Planning and Policy Support for the achievement of Sustainable Development Goals". This is part of the Applied Research Program on Energy for Economic Growth (EEG), led by Oxford Policy Management. The views expressed in this paper do not necessarily reflect the UK government's official policies.

**Acknowledgments:** We would like to sincerely thank Zar wali Habibi and Mahmudshah Sediqi (Da Afghanistan Breshna Sherkat - DABS), Niaz Zaki, Ahmad Rasooli, and Ahmad Shekeb Faiz (Ministry of Rural Rehabilitation and Development - MRRD), Hasibullah Habibzada (Ministry of Energy and Water - MEW), Mohammad Zaher Sultani (Kabul Polytechnic University - KPU), Hamidullah Adina (Kabul University - KU), Hasibullah Samadi (Afghan Geodesy and Cartography Head Office - AGCHO), Ahmad Farid Formuli (Capital Region Development Authority - CRIDA), and Noorullah Stanikzai. This paper would not have been possible without their generous and insightful contributions. None of these individuals should be held responsible for any erroneous facts or interpretations. Any remaining errors are solely the responsibility of the authors.

**Conflicts of Interest:** The authors declare no conflict of interest.

## Appendix A Model Assumptions and Input Parameters

The analysis was conducted based on socio-economic data (1) to determine electricity demand. Spatially explicit energy resources (2) were required to determine the potential for distributed generation. Techno-economic data (3) were collected for all technologies taken into consideration. Finally, spatially explicit fragility related information (4) was added as an extension for this experiment. Values and assumptions have been developed based on literature review and in consultation with local counterparts in Afghanistan (see acknowledgement section). Nevertheless, they are volatile and are only illustrative in the context of this paper.

*Appendix A.1 Socio-Economic Parameters*

The targeted level of electricity consumption per settlement is a major modelling parameter. We choose to focus on electrifying households because of the direct improvement to life quality. Household demand estimates were based on the population projections in combination with an energy-use target (kWh/capita/year) at each settlement. The energy-use is based on the multi-tier framework for energy access developed by the World Bank in 2015 [82]. According to this framework, five access tiers are defined. At the lowest, electricity-use is sufficient for tasks such as turning on a light for a few hours and/or charging a mobile-phone or radio battery. The highest tier provides sufficient supply for energy intensive activities such as the running of a refrigerator, air conditioner, cooking appliance, or other machinery. A distinction between urban and rural settlements was induced because these two groups usually follow slightly different population growth and demand profile patterns [57]. The electrification target for urban settlements was set at Tier 5 (598.6 kWh/capita/year) and for rural at Tier 3 (160.6 kWh/capita/year), as these levels were in line with the established electrification goal in the country [57]. The population characteristics that were used for this analysis are presented in Table A1. Note that 2016 was selected as the base year of the analysis due to lack of more recent publicly available data.

**Table A1.** Population characteristics for urban and rural settings in Afghanistan.

| Parameter | Metric | Value 2016 | Value 2030 |
|---|---|---|---|
| Population, total | Million people | 33.73 [83] | 44.310 (estimated based on growth rates, below) |
| Urban population | Percent of total population | 26.3% [84] | 35.8% (estimated based on growth rates, below) |
| Rural population | Percent of total population | 73.7% [84] | 64.2% (estimated based on growth rates, below) |
| Urban growth | Percent growth per year | 3.96% [84] | 3.49% (average value used in the model as 3.65% per year) |
| Rural growth | Percent growth per year | 1.85% | 1.12% (average value used in the model as 1.35% per year) |
| Modelled electricity access | Percent of total population | 30% [57] | 100% |
| Modelled electricity access, urban | Percent of urban population | 89% [57] | 100% |
| Modelled electricity access, rural | Percent of rural population | 11% [57] | 100% |
| People per household, urban | People per household | 7.4 [85] | 7 (assuming 5% decrease over the 15-year period) [86] |
| People per household, rural | People per household | 8.5 [85] | 8.1 (assuming 5% decrease over the 15-year period) [86] |

*Appendix A.2 "Resource" Mapping*

Using GIS data of mean annual wind speed, as in [87], the yearly expected wind energy production was estimated for each location in the country. Similarly, the average annual global horizontal irradiation (GHI) was used to estimate the annual irradiance in every settlement [58]. Small scale hydropower potential was assessed following the methodology suggested by [88]; it yielded an amount of 664 technically exploitable sites in the country. Finally, generating costs for diesel gensets were assessed by considering transportation costs to reach even the most remote locations; the transportation costs were based on the travel time (in hours) from the closest urban center, as described by [58]. Figures A1 and A2 illustrate resource availability over Afghanistan.

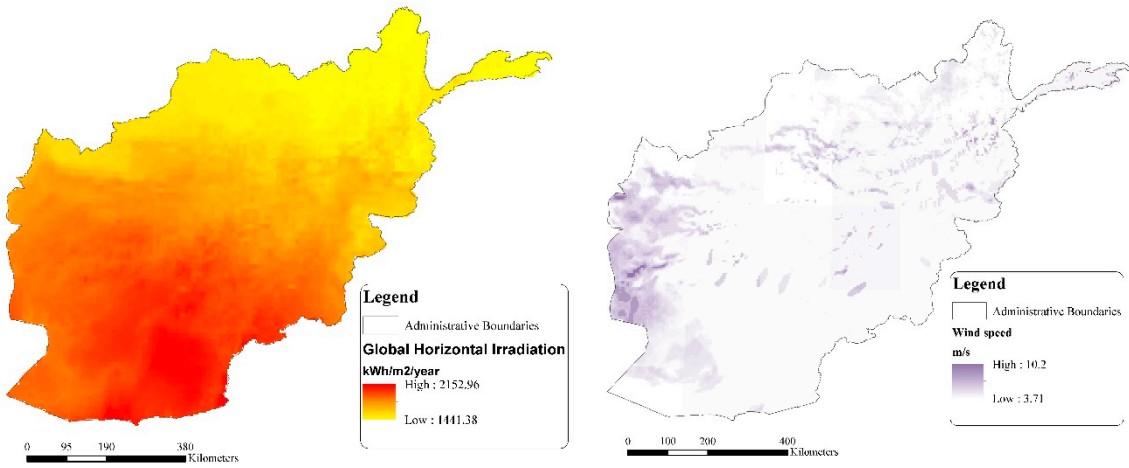

**Figure A1.** Solar and wind resources potential in Afghanistan; described through global horizontal irradiation (kWh/m²/year) and wind speed (m/s at 50 m hub height), respectively.

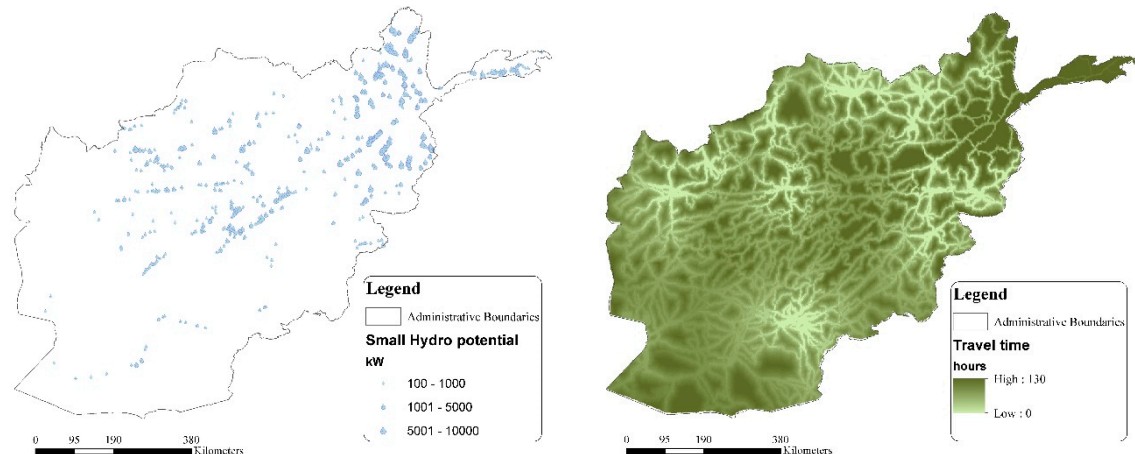

**Figure A2.** Small-scale hydropower potential and travel time in Afghanistan. The first illustrates potential sites ranging from 0.1–10 MW; the latter presents the travelling hours (h) to the nearest town of 50,000 people and is used in order to estimate transportation costs for diesel generators following the methodology presented in [58].

*Appendix A.3 Techno-Economic Parameters*

In terms of electricity supply, we compare three alternative options, namely: (1) grid extension, (2) mini-grids, and (3) stand-alone systems. Each has differing techno-economic characteristics that are briefly presented below. Costs related to the centralized national power grid in Afghanistan were estimated based on the National Supply Energy Plan (2012) [89] and the Power Sector Master Plan [26]). Approximately 3100 MW are planned to be added to the national system by 2025. The plan envisages 2500 MW from 13 hydropower projects, 400 MW from coal power plants in the Aynak and Hajigak mine sites, and 200 MW from the Sheberghan natural gas power plant. Therefore, according to [57], the expected generating cost of electricity for the centralized grid in 2030 is 0.077 USD/kWh; the value has been estimated based on the current development plants. In a similar manner, the expected average cost per additional power unit is estimated at 1970 USD/kW [57]. Other costs related to the extension of the grid are presented in Table A2.

**Table A2.** Parameters related to the extension of the national electricity grid; values adopted by [58].

| Parameter | Cost Unit |
|---|---|
| High-voltage (HV) lines (~110 kV) | 120,000 USD/km |
| Medium-voltage lines (~20 kV) | 9000 USD/km |
| Low-voltage lines (~0.2 kV) | 5000 USD/km |
| MV/LV transformer (50 kVA) | 3500 USD/unit |
| Transmission losses | 18.3% |
| Connection cost per household | 122 USD |
| Cost of generating electricity | 0.077 USD/kWh |
| Capital investment per kW added | 1970 USD/kW |

Note: kV = kilovolts; LV = Low Voltage; MV = Medium Voltage.

In a similar manner, techno-economic characteristics were collected for the off-grid technologies considered in this analysis. The values reflect capital, operating, and fuel costs in Afghanistan as identified in the existing literature [57]. Findings are presented in Table A3. Note that plant capacity is an indicative value to illustrate the typical capacity of each type of system, and efficiency refers to thermal efficiency. The price of diesel in 2030 was estimated based on a projection of the base year value (0.69 USD/liter), considering crude oil price projections from the International Energy Agency [90].

**Table A3.** Electricity generation technology parameters used in the model; values adopted by [58].

| Plant Type | Plant Capacity (kW) | Investment Cost (USD/kW) | O&M Costs (% of Investment Cost/Year) | Fuel Price USD/Liter (Future Value) | Efficiency % | Capacity Factor | Life (Years) |
|---|---|---|---|---|---|---|---|
| **Mini-grid Diesel generator** | 100 | **1200** | 10.0 | 1.00 | 37 | 0.7 | 15 |
| **Mini-grid Small scale hydro** | 1000 | **2500** | 2.0 | - | - | 0.5 | 30 |
| **Mini-grid Solar PV** | 100 | **2600** | 1.8 | - | - | Obtained for each grid point depending on solar availability | 20 |
| **Mini-grid Wind turbine** | 100 | **2300** | 3.5 | - | - | Obtained for each grid point depending on wind availability | 20 |
| **Stand-alone Diesel generator** | 1 | **2000** | 10.0 | 1.00 | 28 | 0.5 | 10 |
| **Stand-alone Solar PV** | 0.4 | **5500** | 1.8 | - | - | Obtained for each grid point depending on solar availability | 15 |

Note: Investment costs include Balance of System (BoS) costs.

Note that, in order to create a proxy for the variation of the risk premiums in Afghanistan, we have used the case of the Sheberghan Combined-Cycle Gas Turbine (CCGT) power plant and the CASA-1000 transmission line. The Sheberghan power plant is located in northern Afghanistan in an area that is characterized by medium fragility. According to the literature, construction costs for the Sheberghan power plant were 60% higher than a comparable unit in neighboring countries (Pakistan/Turkmenistan, both in line with the average international values for the construction costs for CCGT generators) [65]. Similarly, the CASA-1000 transmission line passes through areas with low and/or medium fragility. The literature indicates that construction costs are at least ~18% higher than initial estimates, due to "substantial logistics and security costs" [65].

Finally, the default discount rate in this analysis was set at 12%, based on the calculation of the WACC value and the following assumptions for financing structures, as defined in [65,77,91]:

- Portion of financing: 30% equity–70% debt
- Nominal cost of capital: 12% for equity (DABS)–15% for debt (estimate for high risk projects)
- Tax rate: 0% for equity–20% for debt
- Inflation rate–5% for equity (Afghanistan)–1.5% for debt

*Appendix A.4 Fragility Information*

Finally, in order to incorporate spatial elements of fragility into the analysis, we processed and used a map indicating areas of high and low support to militia groups [92]. The initial imagery was digitized using a geo-referencing process on the Asia Lambert Conformal Conic coordinate system. Then, it was converted into a five class raster layer using supervised classification—Maximum Likelihood Classification Technique—and further processed using filtering and smoothing in order to minimize random noise. The map was projected to the WGS 84/UTM zone 42N system, in accordance with the rest of the layers. Figure A3 illustrates the final product.

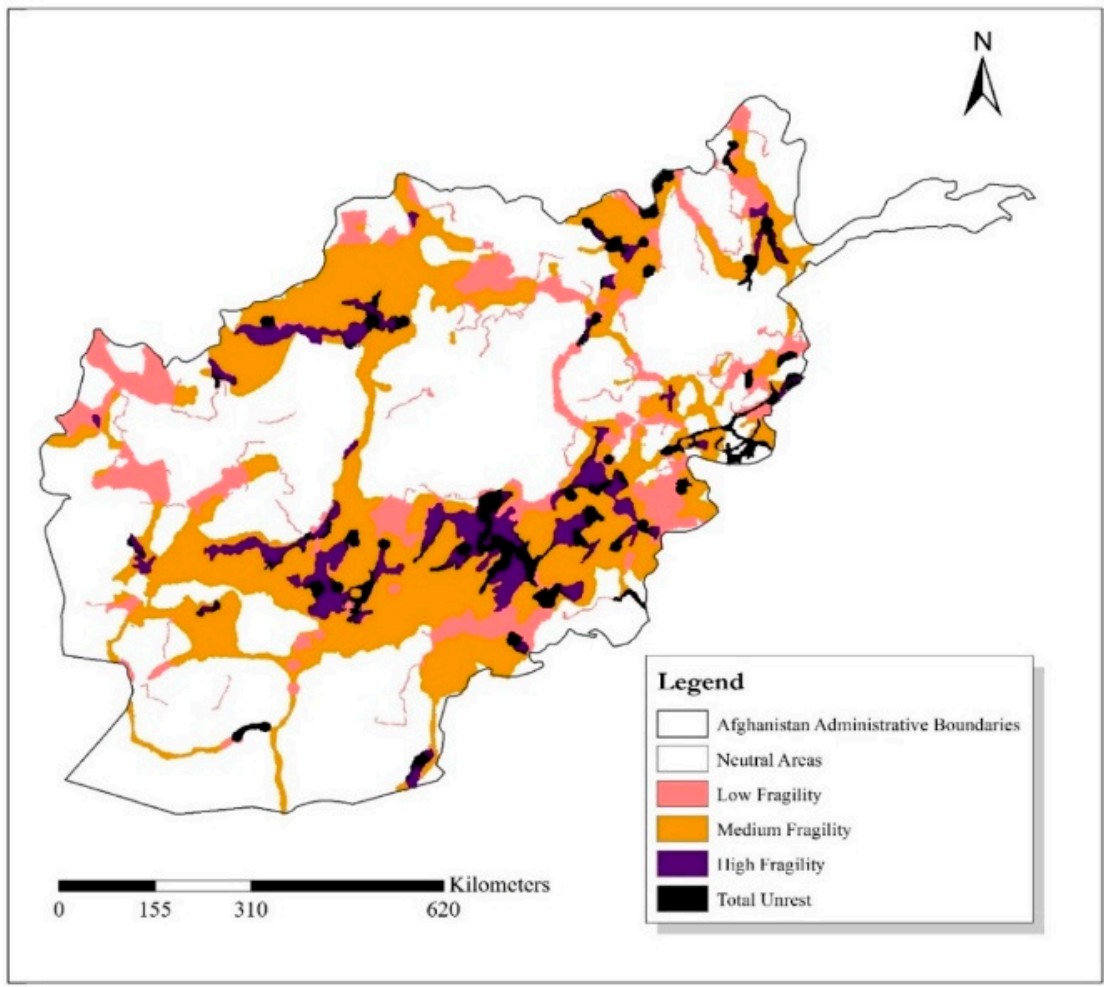

**Figure A3.** Fragility map in Afghanistan in 2016, as adopted by [92]. Areas under militia control are characterized by total unrest, whereas high and low confidence support zones match with high and medium fragility groups from the table above. Areas with no available data or that are neutral are characterized with low fragility.

## Appendix B

**Table A4.** Summaries of electrification results per province.

| Province name | Total Area (sq. km) | Fragile Area (% of Total) | Baseline Scenario | | | | | Scenario A | | | | | Scenario B | | | | | Scenario C | | | | |
|---|---|---|---|---|---|---|---|---|---|---|---|---|---|---|---|---|---|---|---|---|---|---|
| | | | Optimal Technology Mix (people) | | | Investment | Capacity | Optimal Technology Mix (people) | | | Investment | Capacity | Optimal Technology Mix (%) | | | Investment | Capacity | Optimal Technology Mix (%) | | | Investment | Capacity |
| | | | Grid | Mini-grid | Stand-alone | Million USD | MW | Grid | Mini-grid | Stand-alone | Million USD | MW | Grid | Mini-grid | Stand-alone | Million USD | MW | Grid | Mini-grid | Stand-alone | Million USD | MW |
| Nimroz | 21,235 | 32.9% | 60,478 | 60,788 | 100,780 | 86.9 | 15.6 | 60,792 | 59,738 | 101,516 | 84.6 | 15.6 | 60,478 | 51,006 | 110,561 | 88.4 | 15.6 | 60,792 | 50,050 | 111,204 | 85.9 | 15.5 |
| Hilmand | 59,033 | 42.8% | 445,758 | 434,874 | 470,665 | 489.5 | 89.6 | 452,613 | 423,802 | 474,882 | 477.9 | 89.3 | 467,792 | 77,065 | 806,440 | 550.0 | 87.1 | 469,877 | 74,489 | 806,931 | 528.8 | 87.0 |
| Kandahar | 51,135 | 52.3% | 1,365,134 | 459,406 | 470,840 | 818.7 | 215.8 | 1,371,273 | 449,325 | 474,782 | 808.3 | 215.5 | 1,367,341 | 222,097 | 705,942 | 854.4 | 214.6 | 1,373,278 | 215,169 | 706,933 | 838.1 | 214.3 |
| Farah | 61,634 | 40.4% | - | 328,395 | 449,815 | 417.0 | 84.0 | - | 324,164 | 454,045 | 405.5 | 84.0 | - | 228,872 | 549,337 | 459.2 | 83.6 | - | 224,158 | 554,051 | 444.8 | 83.6 |
| Zabul | 22,292 | 91.1% | - | 299,969 | 192,610 | 234.9 | 43.9 | - | 292,697 | 199,882 | 230.3 | 43.9 | - | 6,036 | 486,543 | 276.8 | 42.7 | - | 5,209 | 487,370 | 262.4 | 42.7 |
| Paktika | 15,691 | 83.3% | - | 555,829 | 99,320 | 272.4 | 61.7 | - | 553,292 | 101,857 | 269.9 | 61.7 | - | 207,527 | 447,622 | 357.4 | 60.0 | - | 201,564 | 453,585 | 344.0 | 60.0 |
| Ghazni | 29,241 | 79.6% | 544,026 | 1,311,481 | 117,955 | 838.0 | 195.9 | 574,089 | 1,272,580 | 126,793 | 836.9 | 194.4 | 604,985 | 301,656 | 1,066,821 | 1097.7 | 188.4 | 614,360 | 288,374 | 1,070,728 | 1063.8 | 187.9 |
| Uruzgan | 16,630 | 100.0% | 343 | 442,495 | 109,231 | 244.3 | 49.6 | 343 | 437,319 | 114,407 | 241.4 | 49.5 | 343 | 59,812 | 491,914 | 302.9 | 47.8 | 343 | 56,935 | 494,791 | 288.7 | 47.8 |
| Hirat | 69,648 | 35.0% | 1,573,004 | 1,086,329 | 469,015 | 946.1 | 218.8 | 1,586,794 | 1,066,757 | 474,797 | 936.4 | 218.4 | 1,582,080 | 860,144 | 686,125 | 997.2 | 217.4 | 1,592,779 | 834,471 | 701,098 | 983.0 | 217.0 |
| Khost | 5,158 | 92.6% | - | 897,521 | 1,496 | 494.0 | 156.4 | - | 897,440 | 1,577 | 494.0 | 156.4 | - | 766,651 | 132,366 | 661.2 | 155.8 | - | 762,039 | 136,978 | 657.4 | 155.7 |
| Daykundi | 23,170 | 30.7% | - | 600,446 | 134,500 | 342.3 | 68.9 | - | 586,075 | 148,872 | 340.0 | 68.8 | - | 494,018 | 240,928 | 357.6 | 68.4 | - | 481,268 | 253,679 | 351.8 | 68.3 |
| Paktya | 8,316 | 88.1% | 124,602 | 761,021 | 7,698 | 343.9 | 83.6 | 135,876 | 748,802 | 8,643 | 344.3 | 83.0 | 185,806 | 513,088 | 194,427 | 431.3 | 79.5 | 190,459 | 505,191 | 197,670 | 424.5 | 79.2 |
| Ghor | 43,572 | 20.2% | - | 670,173 | 289,969 | 471.5 | 89.8 | - | 658,888 | 301,254 | 465.2 | 89.7 | - | 566,122 | 394,020 | 486.1 | 89.3 | - | 556,555 | 403,586 | 476.5 | 89.2 |
| Wardak | 11,912 | 42.9% | 118,533 | 631,061 | 25,218 | 322.6 | 73.4 | 173,987 | 574,128 | 26,698 | 322.1 | 70.5 | 246,281 | 317,634 | 210,897 | 389.1 | 63.8 | 253,633 | 312,353 | 208,826 | 382.0 | 63.5 |
| Logar | 4,989 | 69.0% | 319,984 | 301,771 | 976 | 292.3 | 70.4 | 333,544 | 288,062 | 1,125 | 306.5 | 69.7 | 332,413 | 137,735 | 152,582 | 424.1 | 69.1 | 333,073 | 136,162 | 153,496 | 423.6 | 69.0 |
| Nangarhar | 9,627 | 63.0% | 2,415,449 | 323,140 | 3,473 | 921.3 | 217.0 | 2,455,609 | 282,437 | 4,016 | 931.7 | 214.7 | 2,423,481 | 150,223 | 168,358 | 1136.9 | 215.7 | 2,440,259 | 132,464 | 169,339 | 1128.9 | 214.3 |
| Bamyan | 17,724 | 4.1% | - | 447,030 | 90,296 | 254.3 | 52.3 | - | 441,877 | 95,449 | 252.4 | 52.3 | - | 439,794 | 97,532 | 255.7 | 52.3 | - | 434,068 | 103,258 | 253.7 | 52.3 |
| Kabul | 5,758 | 52.0% | 8,308,540 | 76,075 | 4,496 | 1,227.2 | 457.6 | 8,310,416 | 73,741 | 4,954 | 1227.4 | 457.5 | 8,316,874 | 45,227 | 27,009 | 1239.8 | 457.0 | 8,317,734 | 43,873 | 27,504 | 1239.5 | 457.0 |
| Laghman | 4,907 | 40.8% | 467,193 | 194,751 | 2,113 | 187.8 | 38.3 | 478,334 | 183,490 | 2,233 | 188.5 | 37.7 | 499,477 | 122,415 | 42,164 | 208.5 | 36.5 | 510,316 | 111,389 | 42,352 | 207.9 | 35.9 |
| Badghis | 26,591 | 46.1% | - | 538,414 | 148,285 | 339.6 | 77.7 | - | 533,263 | 153,436 | 333.5 | 77.4 | - | 344,520 | 342,178 | 391.9 | 76.7 | - | 338,702 | 347,997 | 375.0 | 76.0 |
| Kunar | 6,191 | 62.6% | 21,372 | 761,495 | 3,022 | 382.9 | 111.8 | 38,344 | 743,984 | 3,562 | 383.2 | 110.8 | 400,944 | 259,101 | 125,845 | 471.8 | 71.6 | 400,089 | 249,886 | 135,914 | 425.0 | 69.8 |
| Parwan | 7,599 | 23.1% | 1,033,928 | 318,558 | 16,794 | 308.5 | 85.1 | 1,064,621 | 286,704 | 17,956 | 308.8 | 83.4 | 1,070,503 | 253,844 | 44,933 | 320.8 | 82.9 | 1,073,631 | 249,302 | 46,348 | 320.1 | 82.7 |
| Kapisa | 2,463 | 61.6% | 600,152 | 80,777 | 4,372 | 250.9 | 62.1 | 605,707 | 74,953 | 4,640 | 252.1 | 61.7 | 626,504 | 26,564 | 32,232 | 275.0 | 60.5 | 631,564 | 23,901 | 29,835 | 275.0 | 60.2 |
| Baghlan | 26,964 | 26.2% | 551,098 | 854,310 | 69,683 | 525.4 | 117.9 | 555,634 | 843,463 | 75,994 | 522.9 | 117.4 | 569,897 | 673,307 | 231,886 | 571.1 | 115.4 | 569,103 | 666,190 | 239,798 | 554.7 | 114.4 |
| Sari Pul | 20,010 | 41.6% | 157,713 | 596,282 | 88,679 | 349.5 | 75.4 | 172,420 | 575,777 | 94,476 | 347.9 | 74.6 | 176,593 | 441,930 | 224,150 | 405.8 | 73.6 | 202,085 | 404,439 | 236,149 | 401.3 | 72.0 |
| Nuristan | 11,499 | 35.3% | 964 | 175,851 | 85,510 | 137.9 | 27.0 | 1,629 | 174,664 | 86,032 | 135.3 | 26.9 | 1,086 | 114,432 | 146,807 | 152.5 | 26.7 | 3,756 | 109,513 | 149,056 | 147.1 | 26.3 |
| Panjshir | 4,504 | 0.0% | 6,112 | 139,640 | 21,784 | 76.1 | 16.6 | 8,480 | 137,037 | 22,019 | 75.5 | 16.5 | 6,112 | 139,640 | 21,784 | 76.1 | 16.6 | 7,272 | 138,245 | 22,019 | 75.5 | 16.6 |
| Faryab | 25,532 | 83.1% | 618,923 | 661,361 | 154,400 | 473.5 | 106.5 | 622,445 | 652,475 | 159,764 | 467.6 | 106.1 | 631,438 | 233,083 | 570,163 | 592.0 | 103.7 | 631,538 | 227,504 | 575,641 | 567.1 | 103.0 |
| Samangan | 14,318 | 30.3% | 74,215 | 417,411 | 38,080 | 242.1 | 50.9 | 77,399 | 410,943 | 41,364 | 240.6 | 50.6 | 86,401 | 298,619 | 144,686 | 268.4 | 49.6 | 90,435 | 290,435 | 148,836 | 264.5 | 49.3 |
| Badakhshan | 43,645 | 34.3% | - | 1,440,411 | 123,023 | 760.5 | 173.9 | - | 1,428,291 | 135,142 | 754.5 | 173.5 | - | 1,190,526 | 372,907 | 840.7 | 172.5 | - | 1,166,489 | 396,944 | 821.9 | 171.5 |
| Balkh | 21,601 | 30.7% | 1,386,421 | 598,604 | 47,722 | 521.6 | 130.4 | 1,406,003 | 566,060 | 60,684 | 516.6 | 128.5 | 1,401,737 | 499,299 | 131,711 | 563.0 | 129.0 | 1,418,115 | 469,253 | 145,380 | 555.0 | 127.2 |
| Takhar | 15,785 | 33.0% | 537,292 | 867,618 | 16,182 | 498.6 | 116.0 | 544,371 | 857,718 | 19,003 | 501.8 | 116.8 | 635,813 | 628,808 | 156,470 | 585.4 | 108.1 | 620,641 | 624,780 | 175,670 | 564.1 | 107.6 |
| Jawzjan | 15,136 | 60.4% | 479,786 | 287,616 | 71,621 | 281.3 | 57.6 | 497,454 | 268,769 | 72,801 | 280.0 | 56.4 | 523,866 | 144,543 | 170,615 | 339.4 | 54.3 | 538,516 | 122,935 | 177,573 | 335.1 | 53.2 |
| Kunduz | 10,276 | 80.8% | 1,347,553 | 178,091 | 22,632 | 292.4 | 62.0 | 1,362,909 | 161,102 | 24,265 | 289.1 | 60.5 | 1,365,469 | 86,677 | 96,129 | 359.2 | 60.2 | 1,363,324 | 80,129 | 104,822 | 343.1 | 58.9 |
| Afghanistan | 733,786 | 46.1% | 22,558,574 | 17,798,994 | 3,952,254 | 14,645.7 | 3553.0 | 22,891,088 | 17,329,814 | 4,088,919 | 14,572.5 | 3533.6 | 23,583,713 | 10,902,017 | 9,824,091 | 16,787.3 | 3445.7 | 23,706,975 | 10,587,483 | 10,015,363 | 16,409.9 | 3429.1 |

**Appendix C**

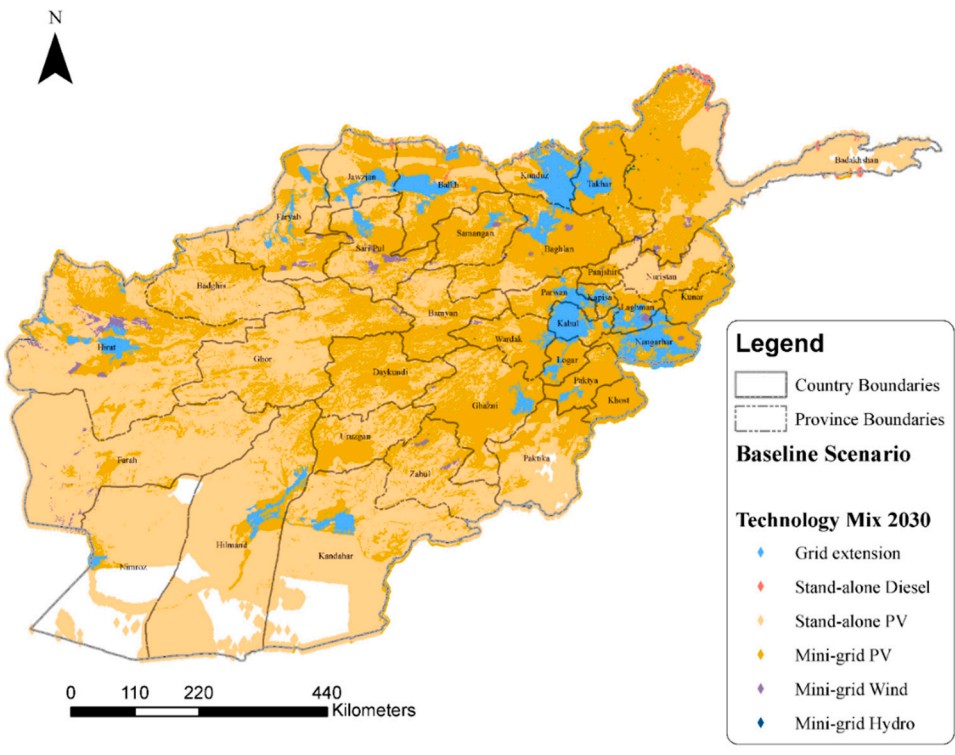

**Figure A4.** Optimal technology mix for universal electrification in Afghanistan by 2030 under the baseline scenario.

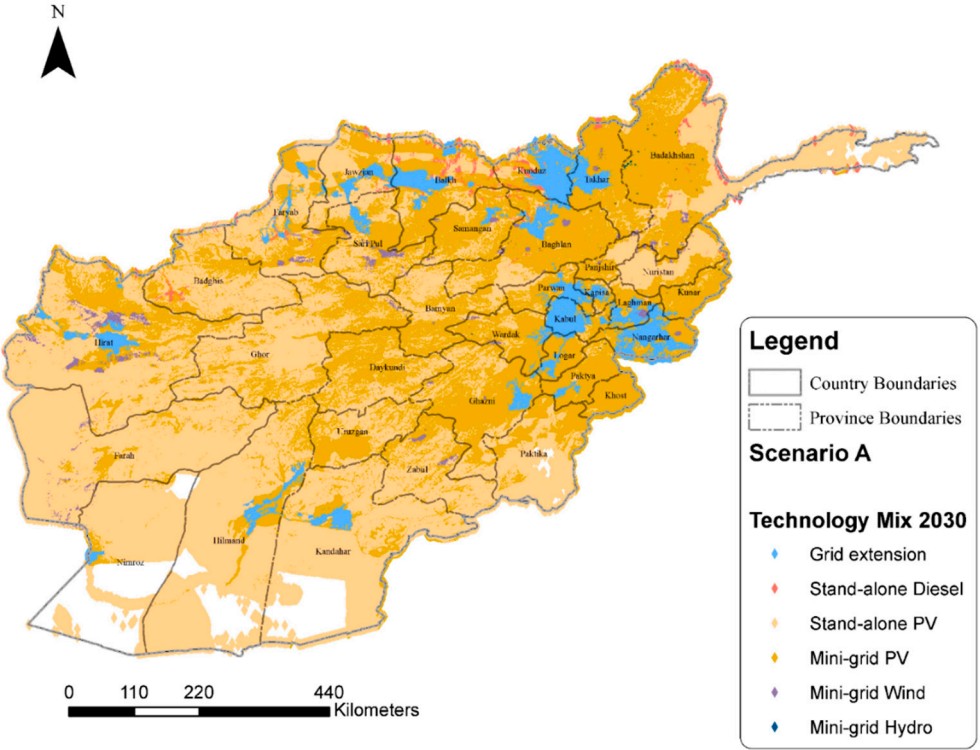

**Figure A5.** Optimal technology mix for universal electrification in Afghanistan by 2030 under scenario A.

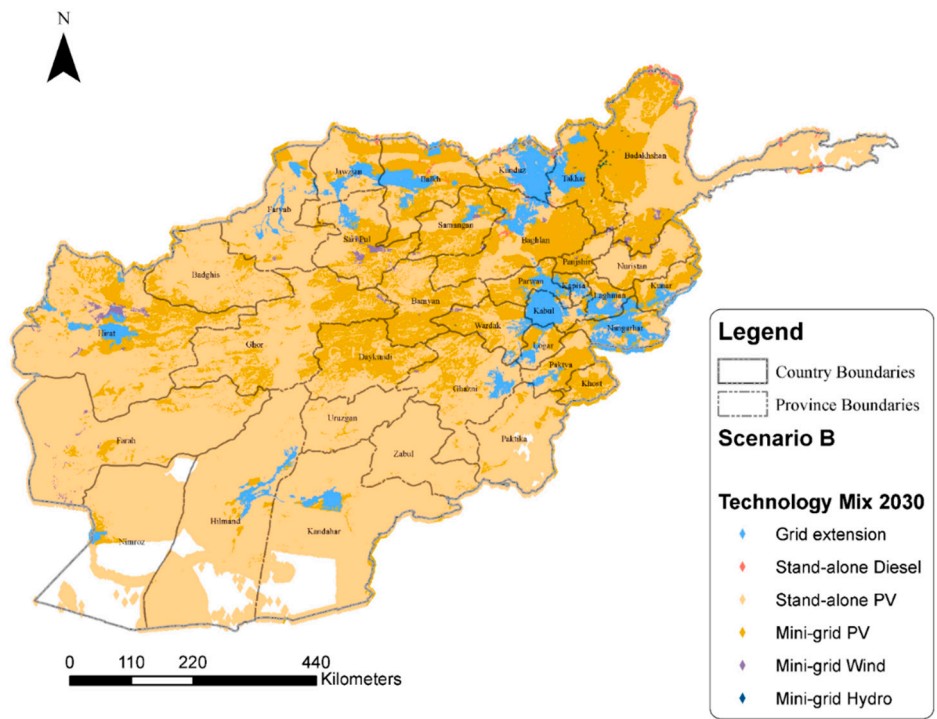

**Figure A6.** Optimal technology mix for universal electrification in Afghanistan by 2030 under scenario B.

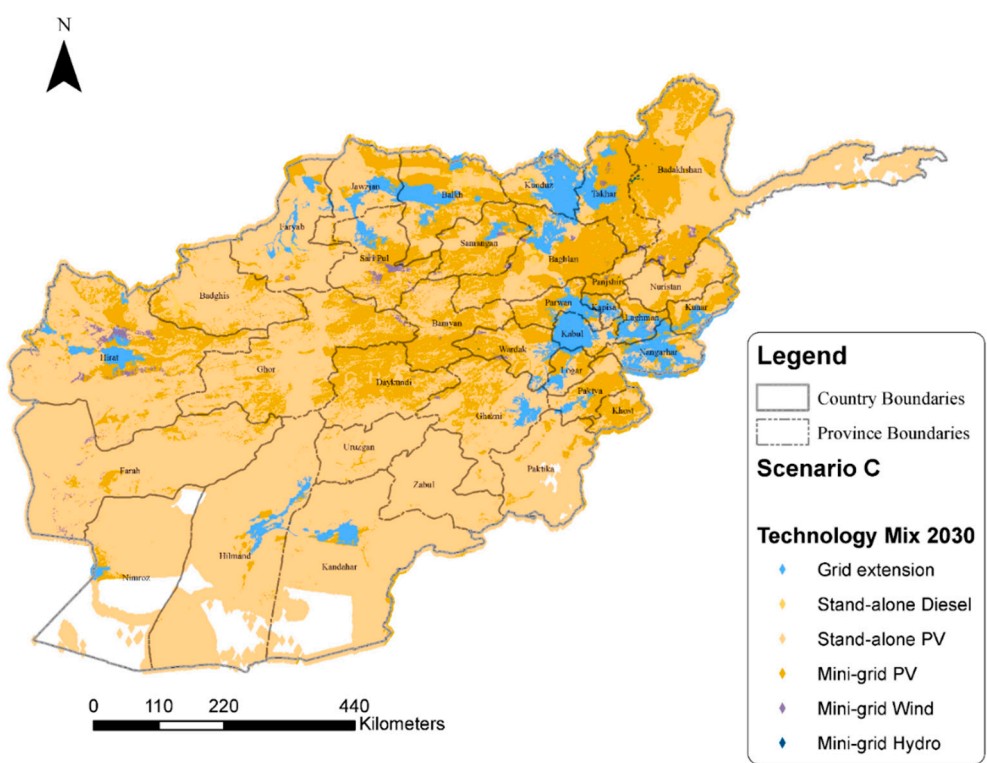

**Figure A7.** Optimal technology mix for universal electrification in Afghanistan by 2030 under scenario C.

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
