# Peer review of "Supporting Electrification Policy in Fragile States: A Conflict-Adjusted Geospatial Least Cost Approach for Afghanistan"

_sustainability, doi:10.3390/su12030777_

Round 1

Reviewer 1 Report

The authors present a new methods how to incorporate conflict into geospatial energy planning and apply it to Afghanistan. They find sensitivity to conflict risk premiums in different settings, with rural areas especially affected by the existence of conflict in terms of the cost-optimal strategy. The subject is interesting and worthy of investigation as too little work has been done in this important space. However, the paper has a few methodological issues (some of which seem to be based on using certain basic scientific terms in a way that did not make immediate sense to me, but I might be just misinterpreting some of the text) which I would recommend be addressed before publication:

Major comments:

Please state your findings in the abstract more clearly: which technology is, on average, best suited to deal with conflict? For me personally, the introduction is too short and does not frame the paper well. It only sets out that there is limited electrification in conflict-ridden states, but does not present the real world issue, nor the academic challenge the authors want to address. Some of this is currently in the background section, but I think these aspects are too general to belong in section 2 – Section 2 should focus much more on past methods in the literature, not to frame the paper I disagree with the authors’ key assumption that risk premiums should only be assigned to on-grid electrification (as they do in section 3), or only to on-grid and mini-grid (as they do in section 4 – not sure actually why the two sections use different assumptions on this key aspect of the model). There are multiple concerns I have with the author’s justification and narrative here: I am not yet fully convinced by using the requirement of interconnectivity as a reason why stand-alone systems have no risk premiums – solar home systems also needs forms of interconnectivity, albeit in the form of sustained value chains. In order for off-grid to work at scale, one needs elaborate sales, training and distribution channels which may be similarly, or even more complicated to establish and maintain than power lines – i.e. there also exists an “extensive web” as the authors call it. Narayan and colleagues show that under ideal dynamic conditions, lead acid batteries in SHSs fail after around 5 years (Narayan et al., 2018), in reality these times can be considerably shorter. They then need to be recycled in order for the toxins not to pollute the environment and then need to be replaced, ideally by the same supplier that originally sold the system. This continuity seems to me like a further source of risk in volatile regions Off-grid are also often limited in their use case, so comparisons with grid-based electricity needs to make sure that the same voltage and reliability levels can be assured. For instance, the authors say that “Once intermittency is accounted for, it compares the lifecycle generation costs for each technology”, but I don’t think this model can account for intermittency – the authors should obviously avoid claiming that their approach can do a certain thing when it is not able to do so. One clear example in this context is that when using a 1 km2 resolution, any voltage drops from household to household which are connected on the same LV line cannot be modelled, albeit it being known to have dramatic effect on the quality of access in mini-grids and grid connections in a developing country context (Jacome et al., 2019) Off-grid is usually driven by the private sector while the public sector focuses on on-grid (Ma and Urpelainen, 2018), so off-grid on the ground usually has a constraint of needing to recover cost (Knuckles, 2016), which is often not the case for on-grid if its state-owned. Few companies are willing to go into conflict zones as profit chances are slim, while the public sector has non-monetary incentives to still do so. It thus may be an interesting theoretical experiment to propose a risk-optimal network where implementation realities on the ground are not considered. However, as the authors aim to “capture the dynamics of electrification in fragile states and provide useful policy insights” the proposed planning model could take into account implementation realities on the ground, i.e. the issue with which planners in developing countries are faced, then depending on the context, de facto conflict risk premiums may indeed be higher for (private) off-grid compared to (public) on-grid solutions. The authors should discuss in some more depth why they chose a method of introducing risk premiums to incorporate political risks into a costing model. There is always the risk of such an effort being a somewhat random allocation of monetary resources or premiums where the subject is not monetary per se. Other previous work incorporates political risk in energy planning models in developing country contexts as an additional objective function rather than as a one-dimensional monetary risk premium (Trotter et al., 2018, 2017). I do not think that any of the two approaches is necessarily better than the other, but it would be useful to give a rational why conflict was dealt with by using assumptions that turn it into a solely monetary measure (in my view a limitation of the study which should at least be mentioned and discussed) I don’t get what the authors mean by “hypothesis” in section 3.2 and 3.5. They never explicitly formulate a hypothesis, so I don’t know what they are testing. In the context of modelling, I am not sure that this is the right word – It seems like what the authors are calling hypothesis are actually assumptions, which is something very different. Modelling results are contingent on assumptions one puts into the model, like for instance those the authors use to capture conflict risks in LCOE calculations. A hypothesis is there to be proved or disproved, but assumptions in models function like axioms and therefore cannot meaningfully be proven. Section 3.5 then says “Testing hypothesis” but it just describes a set of descriptive scenario data rather than any modelling results Just as a question: does the Onsset model incorporate the capital cost of (fracitonal) on-grid generation plants when assigning a cost to “grid extension”? I.e. when an area is electrified, does it look at the additional capital cost of generation? If it doesn’t, then I think this approach underestimates the financial effect conflict has on on-grid electrification. If it does, then it will be difficult to pinpoint where exactly that electricity is coming from, i.e. how to initialise the conflict paramaters in the author’s model The paper lacks detail as to how the authors specified the scenarios with the different numerical values for beta – this is a crucial part of the paper’s assumptions and thus does not belong in the appendix but in the main paper. What I could find in the appendix did not make the rationale clear why exactly these values were chosen for different levels of conflict. Why for instance do mini-grid and grid extension have the same SAFs in section 4 and not in section 3? Why should they be the same? For mini-grids, the power source is close to the households, for on-grid, it may be hundreds of kilometres away – I would imagine that this has different implications for the conflict risk. I am not sure how the authors derive things like closer alignments and harmonisation of donors as a policy consequence from the work they present in their paper – I may have missed it, but I could not identify how this paper addressed donor alignment issues

Minor comments:

Please provide a range for the CPIA score, i.e what is a bad and what is a good score? When talking about the positive impact of rural electrification on development, please cite (Cook, 2011) as well as (Bos et al., 2018). The sentence “first the hypothesis is formulated and tested against Herat and Helmand provinces for proof of concept” only make sense later on when the reader understand that these are the names of two provinces in Afghanistan Please explain in one clear sentence why beta_f is only multiplied with debt finance and not with equity finance Change “poor inaccessibility” to “poor accessibility” Table 3 is confusing, it currently says “T&D lines - - Mini-grids” – please make the labelling clearer to indicate that these values apply to both on-grid and mini-grid

References

Bos, K., Chaplin, D., Mamun, A., 2018. Benefits and challenges of expanding grid electricity in Africa: A review of rigorous evidence on household impacts in developing countries. Energy Sustain. Dev. 44, 64–77.

Cook, P., 2011. Infrastructure, rural electrification and development. Energy Sustain. Dev. 15, 304–313. https://doi.org/10.1016/j.esd.2011.07.008

Jacome, V., Klugman, N., Wolfram, C., Grunfeld, B., Callaway, D., Ray, I., 2019. Power quality and modern energy for all. Proc. Natl. Acad. Sci. 116, 16308–16313.

Knuckles, J., 2016. Business models for mini-grid electricity in base of the pyramid markets. Energy Sustain. Dev. 31, 67–82. https://doi.org/10.1016/j.esd.2015.12.002

Ma, S., Urpelainen, J., 2018. Distributed power generation in national rural electrification plans: An international and comparative evaluation. Energy Res. Soc. Sci. 44, 1–5.

Narayan, N., Papakosta, T., Vega-Garita, V., Qin, Z., Popovic-Gerber, J., Bauer, P., Zeman, M., 2018. Estimating battery lifetimes in Solar Home System design using a practical modelling methodology. Appl. Energy 228, 1629–1639.

Trotter, P.A., Maconachie, R., McManus, M.C., 2018. Solar energy’s potential to mitigate political risks: The case of an optimised Africa-wide network. Energy Policy 117. https://doi.org/10.1016/j.enpol.2018.02.013

Trotter, P.A., Maconachie, R., McManus, M.C., 2017. The impact of political objectives on optimal electricity generation and transmission in the Southern African Power Pool. J. Energy South. Africa 28. https://doi.org/10.17159/2413-3051/2017/v28i3a2451

Reviewer 2 Report

The topic of paper is relevant and topical. The paper deals with conflict-adjusted geospatial power sector least cost planing in Afghanistan.

The authors should follow IMRAD format. Introduction is weak. Please provide references to indicate scientific problem. Please avoid figures in introduction and conclusions. Please indicate strengths and weaknesses of applied approach. Address clearly input and limitation of the study including future research guidelines. Results should be discussed with other studies dealing with least costs power sector planing in fragile states. 

Reviewer 3 Report

Please give full name of GIS when appears first time: Geographic Information Systems (GIS) Could you explain ‘non-state violent attacks’ in more details, in particular the definition. Could you explain the main reasons for attacks on energy infrastructure? Why power networks generally receive higher number of attacks? Could you discuss the measures and actions have been in place to prevent the attacks, what are their effectiveness? Why fragile areas have more difficulties in protecting their energy infrastructure? Could you quote some examples of failures in power infrastructure? What was the estimated economic loss in energy and other dependent sectors in these examples? For traditional centralized grid, apart from high guarding / repairing costs, discussions also need to cover from the load perspective: the power cut is likely to affect wider customers (electric load is also centrally connected to the grid). The decentralized power networks have advantages of be independent, one failure is unlikely to affect other electric load. Figure 2: please state discount rates on x-axis; what are PV MG and PV SA? What is percentage of diesel generators in ‘Other’ category? Do you assume both provinces receive the same level of solar resources? PV normally comes with battery or other energy storage devices. Otherwise if you increase the share of PV, how the electricity will be provided overnight? Therefore, it worth considering the reliability (and capacity discount) in different electrification technologies of energy mix. Although stand-alone PV become favorable when the risks increase, such technology cannot meet electric load overnight. Could you explain the detailed reasons for ‘the economics of grid connection seem to be very little affected by the discount rate’? How did you model or mathematically consider the diesel transportation costs to calculate the share of diesel for both provinces? Figure 4: What does the percentage mean in x-axis? Could you state in the figure? Table 3: Would you consider the Grid & Mini-grids share the equal risk premium for each fragility index? Would main interconnected grid bare higher risks as explained in previous section? What methodology has been used to rank the geospatial regions and obtain fragility index for Afghanistan? As you mentioned the population, the existing grid infrastructure, and the rural / urban areas all have impacts on the electrification mix. How are these factors considered in Figure 8 of Fragility index? For example, for the area with high fragility index, there should be more than one electrification mix driven by these different factors. There are two figure 7? Again PV technologies need to combine with energy storage technologies otherwise the PV cannot meet power supply requirements 24/7 due to its uncertainties and unavailability. When considering investment costs for PV, battery costs also need to be taken into account, while diesel generators or grid have less dependency on battery.

Author Response

Response to reviewer's comments are available in the attached document.

Round 2

Reviewer 1 Report

I would like to thank the authors for assessing my comments and providing a reply to each one of them. Several of my smaller comments have been addressed in a satisfactory manner. Yet the authors choose to mostly ignore my substantive comments, instead just saying that future work can tackle these issues rather than the authors expanding their analyses to address these comments. This type of response is of course unable to alleviate my original concerns with the paper.

In addition, there are some specific issues with four of the substantive comments I made. The comments I made were about:

assigning political risk premiums to on-grid electrification only the comparability between off-grid and on-grid differences in risk appetite of (state-run) on-grid and (private) off-grid electrification location of on-grid plants in the model and associated conflict risks

The specific issues I have with the authors rebuttal on these 4 points are as follows:

The authors say that the risks I mention are different from what they are looking at, but they dont offer an explanation as to why my concerns are not valid, and why it is justified to assume that an off-grid connection, assuming it should have a similar decade-long lifetime as a grid connection, is not vulnerable to conflict and can be assigned a zero risk premium. I am fully aware that the method the authors use is capable of assigning whatever risk premiums they want, but this is not presented in the paper, and there are no clear indications to how this would change the situation. The authors also say that their approach is subjective. I do not believe it is enough for a rigorous journal paper to acknowledge subjectivity, but instead, efforts need to be undertaken to minimise subjectivity to improve rigor. The authors themselves say that a scenario analysis would be a good first step in that direction, but do not provide one in the paper The authors just acknowledge the limitation rather than incorporating ways how to cope with this issue in their analyses Again, the authors acknowledge this point but fail to take it up in their analyses, and are thus unable to mitigate my concern the authors claim that "power plants are considered to be remotely
located and more easily policed" which does not make sense to me: If there is a civil war and one group takes control of an area, do the authors believe that the government will then just position a brigade of soldiers around a dam in that area to allow the engineers to get to work there every day and operate it? The authors are advised to look more closely at the literature, for instance into the case of Liberia during its civil war, or at the issues of stranded generation assets in war zones before making such assertions.

Author Response

(The authors gave the same response as above.)

Reviewer 2 Report

I recommend to publish an article.

Author Response

We thank the reviewer for the positive evaluation of the paper.